# MEgoHand: Multimodal Egocentric Hand-Object Interaction Motion Generation

**Bohan Zhou**[1*]  **Yi Zhan**[1*]  **Zhongbin Zhang**[2]  **Zongqing Lu**[1,3†]
[1] School of Computer Science, Peking University
[2] Department of Automation, Tsinghua University
[3] BeingBeyond

## Abstract

Egocentric hand-object motion generation is crucial for immersive AR/VR and robotic imitation but remains challenging due to unstable viewpoints, self-occlusions, perspective distortion, and noisy ego-motion. Existing methods rely on predefined 3D object priors, limiting generalization to novel objects, which restricts their generalizability to novel objects. Meanwhile, recent multimodal approaches suffer from ambiguous generation from abstract textual cues, intricate pipelines for modeling 3D hand-object correlation, and compounding errors in open-loop prediction. We propose **MEgoHand**, a multimodal framework that synthesizes physically plausible hand-object interactions from egocentric RGB, text, and initial hand pose. MEgoHand introduces a bi-level architecture: a high-level "cerebrum" leverages a vision language model (VLM) to infer motion priors from visual-textual context and a monocular depth estimator for object-agnostic spatial reasoning, while a low-level DiT-based flow-matching policy generates fine-grained trajectories with temporal orthogonal filtering to enhance stability. To address dataset inconsistency, we design a dataset curation paradigm with an Inverse MANO Retargeting Network and Virtual RGB-D Renderer, curating a unified dataset of **3.35M** RGB-D frames, **24K** interactions, and **1.2K** objects. Extensive experiments across **five** in-domain and **two** cross-domain datasets demonstrate the effectiveness of MEgoHand, achieving substantial reductions in wrist translation error (**86.9%**) and joint rotation error (**34.1%**), highlighting its capacity to accurately model fine-grained hand joint structures and generalize robustly across diverse scenarios.

## 1 Introduction

The egocentric perspective is humanity's native mode of interaction, directly reflecting how we perceive and engage with the world [7, 17, 48]. It provides rich contextual cues that are often lost in third-person observations, such as the alignment of gaze and hand motion and the real-time visual-motor feedback that guides manipulation [39]. Generating hand-object motions from first-person views is fundamental for many applications. In AR/VR, it enables precise virtual-real alignment [17, 30, 40, 37, 36], while in robot learning, it facilitates natural imitation from human demonstrations [27, 47, 43, 46]. Despite its significant potential, predicting hand-object interactions from egocentric perspectives remains highly challenging [20]. First, continuous camera motion from head-mounted setups causes unstable and shifting viewpoints, disrupting spatial consistency. Second, frequent self-occlusions by the user's own body often hide the hands or objects, leading to missing visual information. Third, the close distance from the camera introduces strong perspective distortion and rapid scale changes, complicating spatial perception. Finally, distinguishing intentional hand

---

[*]Equal contribution
[†]Corresponding author <zongqing.lu@pku.edu.cn>

39th Conference on Neural Information Processing Systems (NeurIPS 2025).

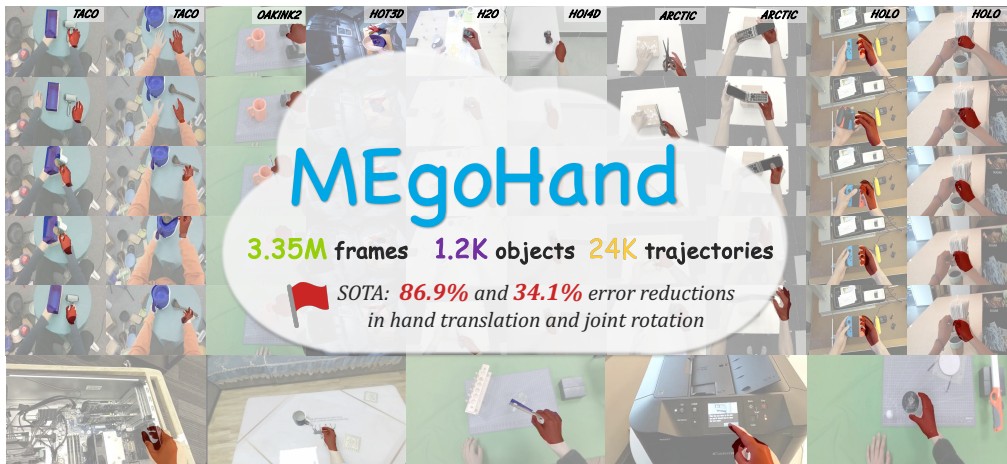

**Figure 1:** MEgoHand stands as the starting point for generating high-quality motion sequences of hand-object interactions, conditioned on egocentric RGB images, textual instructions, and given initial MANO hand parameters.

movements from head-induced ego-motion requires models to reason under partial observations and sparse visual cues.

Existing methods largely ignore the challenges of first-person perspectives because they typically rely on predefined 3D object attributes (*e.g.*, mass, geometry) to model hand-object interactions (HOI) [49, 34, 8, 6]. However, their performance significantly degrades when dealing with novel or unknown objects. Recent efforts have begun to address these limitations of first-person vision and reduce dependence on explicit object models. SIGHT-Fusion [16] mitigates occlusion by segmenting hands and objects in egocentric RGB inputs, but the lack of textual guidance often leads to ambiguous action generation. In contrast, the multimodal LatentAct [31] incorporates visual and textual information along with 3D contact points, enabling more context-aware modeling. Nevertheless, the complex pipeline required for contact map generation limits its feasibility in real-world applications. Furthermore, these methods adopt an open-loop prediction strategy based solely on the first frame. Without corrective feedback, errors caused by viewpoint shifts and occlusions accumulate over time, ultimately leading to cascading failures in interaction prediction.

To address the aforementioned challenges, we propose MEgoHand, a **M**ultimodal **Ego**centric **Hand**-Object Motion Generation approach. Given a textual instruction, an RGB image, and an initial hand pose, MEgoHand synthesizes high-fidelity, physically plausible motion sequences applicable to real-world scenarios. It adopts a bi-level architecture. The High-Level module leverages a vision-language model (VLM) to infer motion priors from visual perception, task understanding, and intent-behavior alignment. Furthermore, to enhance spatial understanding of hand-object relationships without relying on object-specific priors, we incorporate a monocular depth estimator that encodes RGB images into a dense depth representation. The Low-Level module generates fine-grained hand trajectories via a DiT-based flow-matching policy, effectively modeling temporal uncertainty and ensuring motion continuity. To mitigate observation noise induced by egocentric camera motion, the model performs frame-wise prediction of trajectories over the next $l$ frames, followed by Temporal Orthogonal Filtering (TOF) decoding strategy to enhance temporal coherence and stability.

Additionally, we notice that despite the abundance of egocentric hand-object interaction datasets, inconsistencies in language instructions, annotation quality, and pose representations pose significant challenges to unified training. To address these issues, we develop a standardized preprocessing pipeline comprising an Inverse MANO Retargeting Network for pose normalization and a Virtual RGB-D Renderer for generating aligned depth maps. Based on this framework, we curate a multimodal dataset containing **3.35M** RGB-D frames, **24K** interaction trajectories, and **1.2K** objects.

Our experiments show that our MEgoHand achieves SOTA performance across five in-domain and two cross-domain datasets, reducing hand translation and joint rotation errors by 86.9% and 34.1%, respectively. After Procrustes alignment, joint and mesh vertex errors decrease to 0.424 cm and 0.409 cm, corresponding to 71.2% and 71.9% relative improvements over strong baselines. These results

highlight MEgoHand's effectiveness in modeling fine-grained hand joint structures and its strong generalization capability across domains.

The main contributions of this paper are as follows: (1) We propose MEgoHand, the first framework to leverage vision-language models for motion prior inference in egocentric hand-object interactions, augmented with a monocular depth module for object-agnostic spatial reasoning. (2) To address dataset inconsistencies, we design a standardized pipeline with an Inverse MANO Retargeting Network and a Virtual RGB-D Renderer to unify poses and generate aligned depth maps, producing a 3.35M-frame multimodal dataset with 24K interactions and 1.2K objects for unified training and evaluation. (3) MEgoHand outperforms baselines on five in-domain and two cross-domain benchmarks, substantially reducing hand translation and joint rotation errors, demonstrating its effectiveness in fine-grained articulation and robust generalization.

## 2  Related Work

### 2.1  Hand Object Interaction Prediction

Recent advances in hand-object interaction prediction have explored diverse input modalities and generative paradigms to model semantically meaningful and physically plausible behaviors. Object-centric methods such as GEARS [49] and MACS [34] improve motion realism by explicitly modeling physical properties, such as mass and geometry. Similarly, DiffH2O [8] and Text2HOI [6] condition motion prediction on textual descriptions, but still require object-specific information. This reliance on predefined object parameters limits their ability to generalize to novel or unseen objects. In contrast, image-based methods avoid explicit object modeling by learning from visual cues. For example, SIGHT-Fusion [16] predicts hand motion from egocentric images using contact-guided diffusion, demonstrating resilience to occlusions, though it still requires accurate object detection. Moreover, multimodal fusion approaches have shown notable advantages in modeling hand-object interactions. Representative works such as GR00T [4] integrate vision, language, and action modalities for human-like control, while HandsOnVLM [3] leverages textual semantics and visual grounding for hand motion planning. Among them, LatentAct [31] is most relevant to our goal of 3D hand motion generation. It jointly predicts 3D hand trajectories and contact maps from a monocular RGB image, textual action descriptions, and 3D contact points. However, its reliance on an intricately tailored process to generate contact maps hinders its practicality in real-world applications.

### 2.2  Hand Pose Reconstruction

3D hand pose estimation serves as a critical foundation for hand-object interaction prediction by providing accurate 6D pose initialization, enabling physically plausible motion forecasting through spatiotemporal constraints. Current approaches can be broadly classified into image-based and video-based methods. Image-based techniques include HaMeR [28], which scales transformer architectures for pose estimation; Hamba [10], which employs Mamba-based state-space modeling to capture joint dependencies; and SimHand [22], which reduces annotation dependence through contrastive similarity learning. Video-based methods, by contrast, enhance temporal coherence by incorporating motion priors—for example, HMP [11] applies VAE-regularized latent optimization with AMASS priors, while Deformer [14] leverages part-aware deformation-field transformers for dynamic modeling. Beyond isolated hand reconstruction, Hold [13] is the first category-agnostic method for joint articulated hand-object reconstruction from monocular videos, using a compositional implicit model with hand-object constraints.

In addition, egocentric videos, captured from head- or body-mounted cameras, pose distinct challenges for 3D hand pose estimation due to dynamic viewpoints, frequent self-occlusions, and ego-motion artifacts, factors that distinguish them from third-person perspectives. Recent solutions, such as HaWor [45], which employs SLAM-based motion decoupling, and HaPTIC [42], which leverages 4D cross-view attention to ensure temporal consistency, address these difficulties by modeling the complex spatiotemporal dependencies inherent in first-person views. In our study, we focus on egocentric video scenarios where accurate first-frame 3D hand pose estimation serves as the kinematic anchor for subsequent interaction prediction.

# 3 Methodology

## 3.1 Problem Formulation

We represent the hand using the MANO model [32], which is parameterized by hand shape parameters (shape feature $\beta$ and finger rotations $\theta$) and wrist pose parameters (rotation $r$ and translation $t$). To ensure smooth and continuous rotation modeling, we adopt the 6D rotation representation [50] for $\theta$ and $t$. A MANO hand is represented as $h = [\theta; \beta; r; t] \in \mathbb{R}^{109}$, where $\theta \in \mathbb{R}^{15 \times 6}, \beta \in \mathbb{R}^{10}, r \in \mathbb{R}^{1 \times 6}, t \in \mathbb{R}^3$. Given a task description $\mathcal{T}$, visual observation $\mathcal{V}_k$, and the initial $h_k$, the goal is to predict a sequence of future MANO parameter sequences over $l$ time steps. $\mathcal{V}_k$ consists of possible RGB frame $\mathcal{I}_k$ and depth frame $\mathcal{D}_k$.

$$\mathcal{H}_k = \{h_{k+1}, h_{k+2}, \ldots, h_{k+l}\} = \text{MEgoHand}(\mathcal{T}, \mathcal{V}_k, h_k). \tag{1}$$

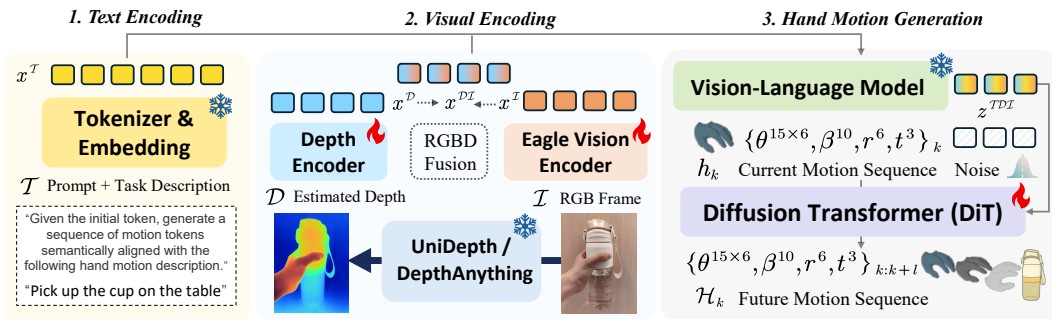

**Figure 2:** During inference, the system prompt and task instruction are encoded using a frozen VLM tokenizer. At each timestep, an RGB image is processed by a pretrained depth estimator to obtain a metric depth map. The RGB and depth images are then combined and encoded into a visual embedding, which—together with the text embedding—is input to the frozen VLM. A DiT-based motion generator receives this multimodal representation along with the initial hand parameters to predict relative future hand motion. During training, the depth encoder, VLM vision encoder, and DiT head are finetuned.

## 3.2 Cerebrum: Multimodal Perception & Understanding

Robust hand motion generation in hand-object interaction demands recognition of target objects in cluttered scenes and reasoning about contact. Traditional approaches depend on explicit object models [49, 34, 8, 6] or intricate modeling of hand-object relationships [31], which can limit adaptability and scalability. In this work, we utilize VLMs for their strong generalizability, enabling automatic extraction of task-relevant semantics and interaction patterns directly from visual observations and textual instructions–without relying on predefined object models. And additionally, to complement the limited 3D spatial understanding of VLMs, we further incorporate a monocular metric depth estimation module that supplies geometric context essential for interaction planning.

**VLM Backbone.** The core of MEgoHand is built upon Eagle-2 [21], a VLM that integrates a SmolLM2 [1] language backbone with a SigLIP-2 [38] vision encoder, both pre-trained on large-scale Internet data. The text tokenizer and transformer blocks are frozen, and the vision encoder is trainable.

**3D Spatial Understanding.** Pre-trained visual encoders are usually effective at 2D semantic understanding while struggling with 3D spatial understanding. To address this, we first incorporate depth into multimodal hand motion generation framework. Specifically, we adopt the monocular metric depth estimator UniDepthV2 [29] to estimate a depth map from an input RGB image. Once the estimated depth map is obtained, we need to encode its spatial features. To the best of our knowledge, there is no existing depth encoder pretrained on large-scale depth datasets. Therefore, we adopt a ResNet-50 [18] encoder pretrained on ImageNet [9]. Although trained on RGB data, we observe that its low-level priors (e.g., edges and textures) transfer effectively to depth inputs. To meet the 3-channel input requirement of ResNet, we replicate the single-channel depth map across channels. To finetune the depth encoder, we use mean squared error (MSE) to align the representation of predicted and ground-truth depth maps. Finally, an additive fusion module combines visual features

$x^{\mathcal{I}}$ and depth features $x^{\mathcal{D}}$ into a unified representation $x^{\mathcal{DI}}$, which interacts with semantic features $x^{\mathcal{T}}$ via cross-modal attention in the Eagle-2 LLM. The resulting output $z^{\mathcal{TDI}}$ captures both the hand-object correlation and the task requirement, enabling holistic understanding and action planning.

### 3.3 Cerebellum: Hand Motion Generation via Flow Matching

**Conditioned Hand Motion Generation.** After the VLM encodes the task instruction $\mathcal{T}$, an RGB frame $\mathcal{I}_k$, and a depth map $\mathcal{D}_k$ into a latent representation $z_k^{TDI}$, a DiT-based motion generator is employed to produce a future motion sequence $\mathcal{H}_k$ of length $l$, a trunk of MANO parameters. This generation is conditioned on the initial MANO parameters $h_k$ because providing an initial hand configuration reduces ambiguity related to hand shape and dexterity, resulting in more realistic and intention-consistent motion sequences. The predicted motion trunk is supervised using a conditional flow matching loss [23, 24]:

$$\mathcal{L}^{\tau}(\theta) = \mathbb{E}_{p(\mathcal{H}_k|h_k, z_k^{TDI}), q(\mathcal{H}_k^{\tau}|\mathcal{H}_k)} \left[ \|\nu_{\theta}(\mathcal{H}_k^{\tau}, h_k, z_k^{TDI}) - \mathbf{u}(\mathcal{H}_k^{\tau}|\mathcal{H}_k)\|^2 \right], \tag{2}$$

where $\mathbf{u}(\mathcal{H}_k^{\tau}|\mathcal{H}_k) = \epsilon - \mathcal{H}_k, \epsilon \sim \mathcal{N}(0, \mathbf{I})$. The subscript $k$ denotes motion timestep, and the superscript $\tau \in [0, 1]$ denotes a flow matching timestep which is sampled from a beta distribution biased toward lower (noisier) values during training. During inference, hand motions are generated by integrating the learned vector field from $\tau = 0$ to $\tau = 1$, starting from Gaussian noise $\mathcal{H}_k^0 \sim \mathcal{N}(0, \mathbf{I})$. The integration follows the forward Euler method:

$$\mathcal{H}_k^{\tau+\delta} = \mathcal{H}_k^{\tau} + \delta \cdot \nu_{\theta}(\mathcal{H}_k^{\tau}, h_k, z_k^{TDI}), \tag{3}$$

where $\delta$ is the integration step size. Please refer to Appendix A.3 for more details. In practice, all transformations are computed in the camera frame so that we can conveniently estimate initial hand using modern detectors [28, 45]. Additionally, we predict the relative wrist transformation to the initial pose and repeat the initial shape parameter $\beta$ as part of the output for each timestep.

**Smooth Decoding Strategy.** To ensure temporal coherence in the generated motion sequences, we propose **Temporal Orthogonal Filtering (TOF)**, a training-free decoding strategy to denoise predicted rotation sequences. At each timestep, we query the motion generator to produce overlapping motion chunks. Let $\hat{R}_k^i, \hat{t}_k^i$ denote the wrist rotation matrix and translation vector at timestep $k$ generated during the query at timestep $i \geq 0$. To suppress high-frequency jitter, a temporal convolution with uniform weights aggregates all rotation and translation estimates corresponding to the same timestep $k$. The resulting translation is given by $\tilde{t}_k = \sum_{t=1}^{l} \hat{t}_k^{k-t}/l$. The resulting convolved rotation $\bar{R}_k$ is then projected onto the closest valid SO(3) manifold via Singular Value Decomposition (SVD), producing the smooth output $\tilde{R}_k$. The process of TOF is formalized in Equation (4). We can freely adjust the frequency of the query to balance inference speed and generation quality.

$$\tilde{R}_k = \underset{R \in \mathrm{SO}(3)}{\arg\min} \left\| R - \bar{R}_k \right\|_F = UV^{\top}, \quad \text{where } USV^{\top} = \mathrm{SVD}(\bar{R}_k), \bar{R}_k = \frac{1}{l} \sum_{t=1}^{l} \hat{R}_k^{k-t} \tag{4}$$

## 4 Datasets Integration

Despite the abundance of egocentric hand-object interaction datasets, inconsistencies in language instructions, annotation quality, and hand pose representations hinder unified training. To address these discrepancies, we systematically integrate and preprocess large-scale public datasets into a unified and standardized training corpus.

**Inverse MANO Retargeting:** Some early datasets, such as FPHA [15], only provide 3D hand joint positions $j^{21 \times 3}$ captured using wearable sensors instead of MANO parameters. The world-frame coordinates of 21 hand keypoints correspond to the output of the MANO model and cannot be directly used as inputs for MEgoHand or as supervision signals for motion generation. To utilize these datasets, we introduce an **Inverse MANO Retargeting Network** $\phi$, which recovers the MANO parameters from joint coordinates. A straightforward approach would be to employ end-to-end supervised learning. However, this method fails completely in practice, as even minor deviations in hand shape—particularly for the shape vector $\beta$—can lead to severely distorted hand geometry and huge reconstruction errors. To overcome this, we propose a novel two-stage iterative training strategy, along with a self-reconstruction loss. As shown in Figure 3a, we first prioritize optimizing the hand

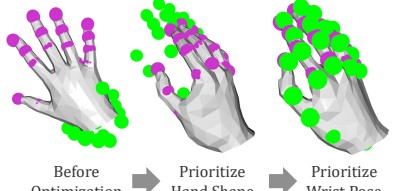

Before Optimization → Prioritize Hand Shape → Prioritize Wrist Pose

**(a)** Green dots denote the ground-truth hand joints, while purple dots represent the joint positions obtained by retargeting hand shape and wrist pose using our Inverse MANO Retargeting Network and reconstructing them via the MANO model.

**(b)** Summary of curated HOI datasets. We split the first 6 for training and the last two for testing.

| Dataset | Frame | Trajectory | Object | Mesh | RGBD |
|---|---|---|---|---|---|
| OakInk2 [44] | 600K | 2.5K | 75 | ✓ | ✗ |
| HOT3D [2] | 400K | 3K | 33 | ✓ | ✗ |
| HOI4D [26] | 400K | 3K | 800 | ✓ | ✓ |
| TACO [25] | 300K | 2.2K | 218 | ✓ | ✓ |
| H2O [19] | 100K | 1K | 8 | ✓ | ✓ |
| FPHA [15] | 100K | 1.3K | 26 | ✓ | ✓ |
| ARCTIC [12] | 250K | 1K | 12 | ✓ | ✗ |
| HOLO [39] | 1200K | 10K | 40 | ✗ | ✓ |
| **Total** | **3.35M** | **24K** | **1.2K** | | |

shape, and upon convergence, shift focus to refining the wrist pose. A gating signal $\sigma$ is used to switch between the two loss functions $\mathcal{L}_1$ and $\mathcal{L}_2$, where $\sigma = 1$ indicates that $\mathcal{L}_1$ has converged, and 0 otherwise. The objective is in Equation (5):

$$\mathcal{L}_1 = w_1 \mathcal{L}_{\text{shape}}\big(\phi(j), \theta, \beta\big) + \mathcal{L}_{\text{recon}}\big(\text{MANO}(\phi(j)), j\big),$$
$$\mathcal{L}_2 = w_2 \mathcal{L}_{\text{pose}}\big(\phi(j), r, t\big) + \mathcal{L}_{\text{recon}}\big(\text{MANO}(\phi(j)), j\big), \qquad (5)$$
$$\mathcal{L}_{\text{inv}} = \sigma \mathcal{L}_1 + (1 - \sigma)\mathcal{L}_2.$$

We pretrain the $\phi$ using 10K paired samples of MANO parameters and joint coordinates covering diverse hand motions, sourced from TACO [25] and OakInk2 [44]. This enables us to reuse the Inverse MANO Retargeting Network to annotate datasets that only provide hand joint positions. For more details on the pretraining process, please refer to Appendix A.2.

**Virtual RGB-D Rendering.** Effective training of the depth encoder requires paired RGB-D data captured from the same egocentric viewpoint. However, many existing datasets, such as ARCTIC [12], HOT3D [2], and OakInk2 [44], provide only RGB image sequences without corresponding depth maps. To address this limitation, we design a **Virtual RGB-D Renderer** to synthesize depth images aligned with the available RGB frames. Given the intrinsic matrix $K$, extrinsic transformation $T_{cw}$, and object points $P_w \in \mathbb{R}^{N \times 4}$ in homogeneous world coordinates, we render the depth map $D$ in the camera view by first transforming points to the camera frame: $P_c = T_{cw} \cdot P_w^T$, then projecting them to pixel coordinates via $p_{uv} = \pi\left(K \cdot (P_c \oslash Z_c)\right)$, where $Z_c = (P_c)z$ and $\pi(\cdot)$ denotes taking the first two components and rounding to the nearest integers. For each pixel $(u, v) = puv^{(i)}$ within the image bounds and with $Z_c^{(i)} > 0$, the depth map is updated as $D[v, u] = \min(D[v, u], Z_c^{(i)})$ to retain the closest surface. We render depth maps for both objects and hands (if visible), capturing accurate hand-object spatial relationships, which are shown to be important in Section 5.

By integrating inverse MANO retargeting and virtual RGB-D rendering, we curate a unified multimodal dataset consisting of **3.35M** RGB-D frames, **24K** interaction trajectories, covering **1.2K** objects. We only consider right hand motion in this submission.

## 5 Experiments

### 5.1 Experiment Setups

**Datasets.** We include 6 training datasets: **TACO**, **FPHA**, **HOI4D**, **H2O**, **HOT3D**, and **OakInk2**. Among them, FPHA was re-annotated using the inverse MANO network and is exclusive for evaluation. For the other five datasets, we hold out 10% of the data from each as in-domain evaluation sets, ensuring no overlap with the training data in terms of action or object categories. Additionally, to assess generalization to unseen domains, we evaluate on two cross-domain test sets: full ARCTIC dataset and a 10% partition of the HOLO dataset as mentioned in [31].

**Metrics.** (1) MPJPE [31]: Mean Per Joint Position Error, the average Euclidean distance between predicted and ground-truth 3D hand joint positions over all timesteps. (2) MPJPE-PA [31]: Procrustes Aligned MPJPE, the MPJPE after applying a single transformation (scale, rotation, translation) to align the predicted and ground-truth joint trajectories. (3) MPVE: Mean Per Vertex Error, the average Euclidean distance between predicted and ground-truth mesh vertices of the MANO model. (4)

MPVE-PA: Procrustes Aligned MPVE, the MPVE after applying Procrustes alignment to remove global scale and pose differences. (5) MWTE: Mean Wrist Translation Error, the average Euclidean distance between predicted and ground-truth 3D wrist translation vectors. (6) MRE: Mean Rotation Error, the average angular difference between predicted and ground-truth joint rotations for $\theta$ and $r$. Given rotation matrices $R_1, R_2 \in \mathbb{R}^{16 \times 3 \times 3}$ for all 16 joints (1 wrist and 15 finger joints), MRE is defined as MRE $= \frac{1}{16} \sum_{j=1}^{16} \cos^{-1}[(\text{trace}(R_{1,j}^T R_{2,j}) - 1)/2]$, where each $R_{1,j}, R_{2,j} \in \mathbb{R}^{3 \times 3}$ represents the rotation matrix of the $j$-th joint. MRE provides a continuous measure of rotational error within the range $[0, \pi]$ radians.

**Baselines.** Among existing approaches, object-centric representation methods [6, 8] are not applicable in our setting because **we do not have access to 3D object models**. Similarly, hand-centric representations [35, 41] are unsuitable since the hand is not always visible. Recent work, **LatentAct** [31], which predicts 3D hand poses and contact maps from a textual action description, a single RGB image, and 3D hand-object contact points, serves as a strong baseline for our task. It does not rely on object models and can operate even when the hand is not visible in the input image. In comparison, MEgoHand takes only a text description and visual observation as inputs. To evaluate the effectiveness of our approach, we compare Transformer-based LatentAct and its diffusion-based variant LatentAct-Diff. Furthermore, to analyze different usages of 3D information, we also compare variants of LatentAct without contact maps and MEgoHand without depth input.

**Modalities.** To comprehensively evaluate the effects of incorporating different modalities, we explore several input configurations: (1) MEgoHand-T only takes textual descriptions; (2) MEgoHand-I only takes RGB images; (3) MEgoHand-ID takes RGB images with depth estimation; (4) MEgoHand-TI takes text and RGB images; (5) **MEgoHand (ours)** incorporates text, RGB images, and depth maps predicted by a foundation depth estimator.

## 5.2 Evaluation on In-Domain Datasets

**Table 1:** Average metrics of in-domain evaluation across 5 datasets: TACO, HOI4D, H2O, HOT3D, and OakInk2. The unit for MRE is radians, and the remaining metrics are measured in centimeters.

| Method | MPJPE↓ | MPJPE-PA↓ | MPVE↓ | MPVE-PA↓ | MWTE↓ | MRE↓ |
|---|---|---|---|---|---|---|
| LatentAct | 7.726 | 1.478 | 7.696 | 1.453 | 7.221 | 0.937 |
| – no concat map | 8.523 | 1.481 | 8.476 | 1.464 | 7.813 | 0.947 |
| LatentAct-Diff | 7.819 | 1.498 | 7.787 | 1.483 | 7.322 | 0.941 |
| – no concat map | 8.802 | 1.582 | 8.752 | 1.564 | 8.051 | 0.950 |
| MEgoHand-T | 8.328 | 0.477 | 8.282 | 0.460 | 7.637 | 0.145 |
| MEgoHand-I | 6.269 | 0.480 | 6.120 | 0.457 | 5.521 | 0.143 |
| MEgoHand-ID | 5.969 | 0.470 | 5.920 | 0.453 | 5.213 | 0.137 |
| MEgoHand-TI | 5.683 | 0.476 | 5.632 | 0.459 | 4.889 | 0.136 |
| **MEgoHand (ours)** | **5.425** | **0.425** | **5.381** | **0.409** | **4.756** | **0.123** |

**Evaluation against Baselines.** As shown in Table 1, our method consistently and significantly outperforms the baseline across all evaluation metrics. Notably, it achieves an 86.9% reduction in mean MRE, resulting in an average rotational deviation of approximately 7 degrees. In contrast, LatentAct struggles to generate accurate finger joints, likely due to its reliance on a single-view RGB image and a single 3D contact point—constraints that severely limit its ability to model the intricate 3D hand-object relationship. Our approach addresses these limitations by: (1) leveraging VLMs for richer contextual understanding, (2) integrating 3D depth estimation to better capture hand-object contact features, and (3) employing closed-loop prediction with TOF decoding strategy to ensure temporally consistent and stable forecasting. Notably, after applying global Procrustes alignment, our method achieves further reductions in joint (MPJPE-PA) and mesh vertex (MPVE-PA) errors to 0.424 and 0.409, corresponding to relative improvements of 71.2% and 71.9% over LatentAct, respectively. These results demonstrate the superior capability of our approach in modeling hand morphology and fine-grained pose prediction.

**Evaluating the Modality Flexibility of Our Model.** We analyze the performance of four variants: text instructions alone, RGB images alone, RGB images + predicted depth maps, text instructions + RGB images. As shown in the green section of Table 1, text-only inputs produce the weakest results.

The absence of visual guidance increases translation error by 61% compared to MEgoHand. RGB-only inputs mitigate this but suffer from ambiguous action patterns, converging to average behaviors due to insufficient 3D spatial cues. MEgoHand-ID resolves these limitations by integrating depth maps, enhancing 3D spatial reasoning, and improving all metrics, achieving a 50% lower MPJPE than LatentAct. To investigate the approaches of incorporating 3D information, we evaluate LatentAct without its contact map against our MEgoHand-TI variant. Crucially, removing the contact map from LatentAct degrades its performance (10.3% MPJPE increase), while MEgoHand-TI achieves a 50% lower error than LatentAct. This indicates that LatentAct critically depends on contact maps, making it less practical for real-world scenarios.

**Evaluation from different datasets.** As illustrated in Figure 4, MEgoHand displays a performance hierarchy: it achieves the strongest results on H2O and OAKINK2, with MPJPE deviations constrained to approximately 3 cm, while exhibiting the weakest performance on HOI4D. We analyze that HOI4D's 800 object instances across 610 scenes account for its generalization challenge. H2O and OAKINK2 feature structured tasks with consistent interaction regions, such as handles and edges, where stable spatial correlations can be learned to enhance generation.

### 5.3 Zero-Shot Transfer on Cross-domain Datasets

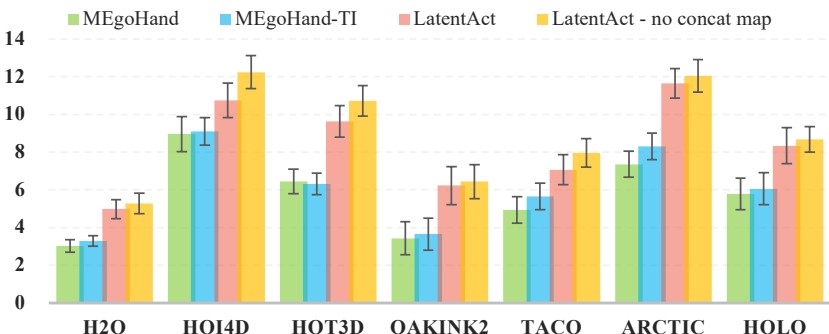

**Figure 4:** The evaluation of our two methods and two baseline variants on five in-domain (H2O, HOI4D, HOT3D, OAKINK2, TACO) and two cross-domain datasets (ARCTIC, HOLO), using MPJPE as metric (unit: cm, lower is better).

To evaluate our method's generalization capacity, we conduct a comprehensive analysis of its zero-shot transfer performance across two cross-domain datasets, spinning object diversity, task complexity, and scene changes. ARCTIC poses greater challenges through complex dynamic coupling between articulated objects and hand configurations (e.g., scissor-cutting requiring coordinated finger-blade kinematics), exposing limitations from our rigid-object training data. Conversely, HOLO features clearer task segmentation and semantically grounded instructions (e.g., "rotate the screwdriver counterclockwise"), which help narrow the action search space and partially mitigate domain shift. Notably, from Table 2, our method achieves SOTA performance with 33.9% and 29.8% MPJPE improvements over the strongest baselines on ARCTIC and HOLO, respectively, demonstrating robust cross-domain transfer capabilities in HOI modeling.

### 5.4 Ablations around Depth

**a) Does a pretrained depth estimator make a difference?** From Table 3, MEgoHand is compatible with various pretrained modern depth estimators like DepthAnythingV2 and UniDepth, achieving comparable results across metrics. This suggests free plug-and-play use of diverse depth estimators.

**b) Do we need real depth supervision?** Comparative analysis with the depth-supervision-ablated variant, here the depth encoder updates solely through motion prediction loss, reveals significant performance degradation in Table 3. This empirically underscores the insufficiency of final motion loss alone to learn spatial-aware representations, necessitating real depth supervision.

**c) Metric depth or relative depth?** We also compare accurate metric depth and relative depth inputs for MEgoHand. The results vary in Table 3. We observe in in-domain scenarios, metric depth can provide consistent scale and distance cues, enhancing 3D spatial understanding. However, in

**Table 2:** Average metrics of out-of-domain evaluation across 2 datasets: ARCTIC and HOLO. The unit for MRE is radians, and the remaining metrics are measured in centimeters.

| Dataset | Method | MPJPE↓ | MPJPE-PA↓ | MPVE↓ | MPVE-PA↓ | MWTE↓ | MRE↓ |
|---|---|---|---|---|---|---|---|
| ARCTIC | LatentAct | 11.65 | 1.975 | 11.58 | 1.942 | 9.920 | 1.577 |
| | – no concat map | 12.04 | 2.023 | 11.96 | 1.990 | 10.25 | 1.590 |
| | LatentAct-Diff | 10.98 | 1.905 | 10.90 | 1.870 | 9.642 | 1.543 |
| | – no concat map | 12.27 | 2.033 | 12.19 | 1.999 | 10.83 | 1.559 |
| | MEgoHand-T | 10.17 | 1.318 | 10.10 | 1.306 | 8.872 | 0.489 |
| | MEgoHand-I | 8.964 | 1.218 | 8.826 | 1.204 | 6.985 | 0.456 |
| | MEgoHand-ID | 8.316 | **1.161** | 8.226 | 1.144 | 6.689 | 0.384 |
| | MEgoHand-TI | 8.305 | 1.173 | 8.194 | 1.126 | 6.313 | 0.452 |
| | **MEgoHand (ours)** | **7.358** | **1.161** | **7.268** | **1.106** | **5.958** | **0.398** |
| HOLO | LatentAct | 8.341 | 1.629 | 8.303 | 1.606 | 8.051 | 1.112 |
| | – no concat map | 8.682 | 1.658 | 8.650 | 1.635 | 8.303 | 1.133 |
| | LatentAct-Diff | 8.235 | 1.605 | 8.196 | 1.582 | 7.973 | 1.101 |
| | – no concat map | 8.492 | 1.631 | 8.453 | 1.609 | 8.172 | 1.118 |
| | MEgoHand-T | 8.605 | 0.879 | 8.572 | 0.860 | 8.204 | 0.499 |
| | MEgoHand-I | 7.525 | 0.841 | 7.484 | 0.819 | 6.871 | 0.416 |
| | MEgoHand-ID | 6.525 | 0.812 | 6.484 | 0.790 | 5.871 | 0.321 |
| | MEgoHand-TI | 6.054 | 0.772 | 6.011 | 0.750 | 5.485 | 0.298 |
| | **MEgoHand (ours)** | **5.775** | **0.697** | **5.747** | **0.673** | **5.437** | **0.271** |

cross-domain scenarios, metric depth is more sensitive to variations in drastic camera parameters alternation, leading to a performance drop.

**Table 3:** The ablation studies of MEgoHand variants across evaluation datasets and test datasets.

| Dataset | Method | MPJPE↓ | MPJPE-PA↓ | MPVE↓ | MPVE-PA↓ | MWTE↓ | MRE↓ |
|---|---|---|---|---|---|---|---|
| Evaluation Datasets | **MEgoHand** | **5.425** | **0.425** | **5.381** | **0.409** | **4.756** | **0.123** |
| | – depthanythingv2 | 5.671 | 0.475 | 5.621 | 0.457 | 4.895 | 0.137 |
| | – no depth supervision | 5.725 | 0.492 | 5.671 | 0.473 | 4.900 | 0.142 |
| | – relative depth | 5.610 | 0.444 | 5.564 | 0.427 | 4.895 | 0.128 |
| ARCTIC | **MEgoHand** | **7.358** | 1.161 | **7.268** | 1.106 | **5.958** | **0.398** |
| | – depthanythingv2 | 8.240 | 1.220 | 8.141 | 1.203 | 6.287 | 0.544 |
| | – no depth supervision | 8.174 | 1.140 | 8.092 | 1.092 | 6.608 | 0.436 |
| | – relative depth | 7.564 | **1.121** | 7.485 | **1.091** | 6.082 | 0.473 |
| HOLO | **MEgoHand** | **5.775** | 0.697 | **5.747** | 0.673 | 5.437 | **0.271** |
| | – depthanythingv2 | 6.094 | 0.895 | 6.055 | 0.873 | 5.512 | 0.331 |
| | – no depth supervision | 6.434 | 0.835 | 6.397 | 0.837 | 5.889 | 0.473 |
| | – relative depth | 5.879 | **0.663** | 5.841 | **0.643** | **5.418** | 0.280 |

## 5.5 Visualization

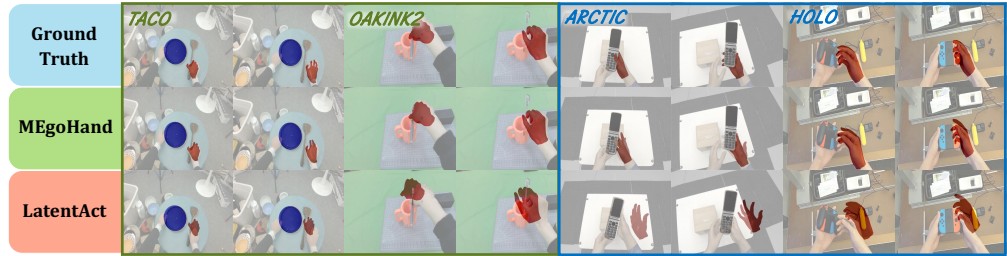

**Figure 5:** We present visualizations across in-domain (green) and cross-domain (blue) datasets. The misalignments of ground-truth annotations are attributed to labeling noise and camera calibration errors. For fair comparison with LatentAct, we provide the initial hand pose and align the motion predictions of LatentAct to the first frame in a chunk.

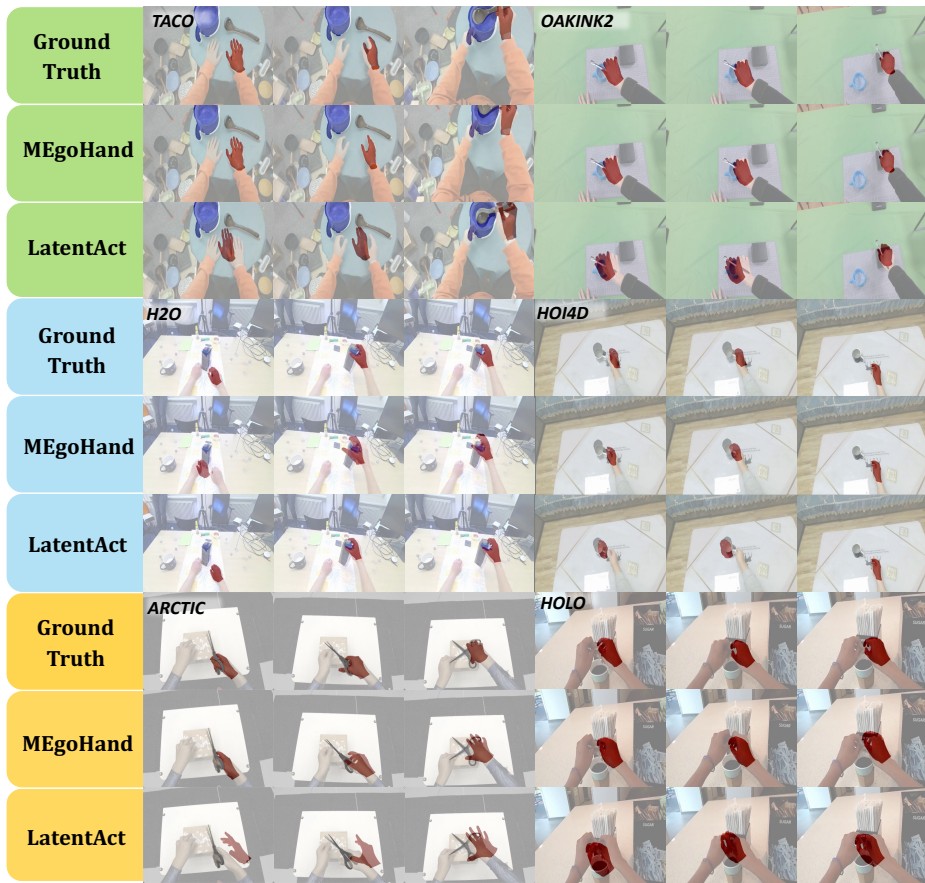

**Figure 6:** Additional visualizations of LatentAct and MEgoHand. Green part is sampled from training sets. Blue part is sampled from evaluation sets. The Yellow part is sampled from testing sets.

We decode the generated MANO parameters into hand mesh vertices and visualize the projections overlaid on the original RGB videos. As illustrated in Figure 5, MEgoHand consistently outperforms LatentAct with more accurate hand poses and finer geometric alignment, particularly in wrist pose and finger joint rotations. We analyze that metric depth inputs play an important role in the generation of higher precision. Besides, we observe that if no hand is visible in the initial frame, LatentAct struggles to predict precise shape parameters $\beta$. This emphasizes the significance of initial hand parameters. Please refer to Appendix B.2 for more visualizations.

## 6 Conclusion & Limitation

We introduce MEgoHand, a multimodal framework for egocentric hand motion generation that integrates initial hand parameters, textual instructions, and RGB images to predict realistic hand-object interaction motion sequences. The hierarchical design combines a vision-language model and depth estimation for semantic understanding and 3D reasoning. A DiT-based motion generator conducts closed-loop prediction, enhanced by Temporal Orthogonal Filtering for temporal consistency. To address data scarcity, we curate a million-scale HOI dataset by leveraging inverse MANO retargeting and virtual RGB-D rendering. As an initial attempt to unify vision language models with 3D reasoning for motion generation, MEgoHand demonstrates strong generalization, achieving SOTA results on five in-domain and two cross-domain datasets.

**Limitation**. Some limitations can be addressed in future research. Utilizing our pretrained inverse MANO retargeting network to annotate a broader range of HOI datasets or adopting modern hand pose detectors [28, 45] to label in-the-wild human videos can further improve data scale, which is promising towards better results.

## Acknowledgements

This work was supported by NSFC under Grant 62450001 and 62476008. The authors would like to thank the reviewers for their valuable comments and advice.

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

# Appendices

## A  Implementation Details

### A.1  Hyperparameters

We report important hyperparameters used for MEgoHand training in Table 1.

**Table 1:** Hyperparameters of MEgoHand Training.

| Hyperparameter | Value |
|---|---|
| Prediction Trunk Size $l$ | 16 |
| Integration Step Size $\delta$ | 0.1 |
| Gradient steps | 50,000 |
| Batch size | 64 |
| Learning Rate | 3e-4 |
| Optimizer | AdamW |
| Adam $\beta_1$ | 0.95 |
| Adam $\beta_2$ | 0.999 |
| Adam $\epsilon$ | 1e-8 |
| LR scheduler | cosine |
| Weight Decay | 1e-5 |
| Warmup Ratio | 0.05 |
| VLM text tokenizer | frozen |
| VLM vision encoder | unfrozen |
| DiT | unfrozen |

### A.2  Inverse MANO Pretraining

**Architecture**. The model architecture of the Inverse MANO Retargeting Network consists of PointNet encoder with 3-layer MLPs.

**Training Parameters**. We set $w_1 = 4.0$ and $w_2 = 5.0$ for $\mathcal{L}_1$ and $\mathcal{L}_2$ respectively. $\mathcal{L}_{\text{shape}}$ and $\mathcal{L}_{\text{recon}}$ are both L1 loss, supervising shape feature $\beta$, translation $t$ or rotation in 6D representation $\theta, r$.

**Visualization**. Figure 1 shows using Inverse MANO Retargeting Network $\phi$ to label FPHA dataset.

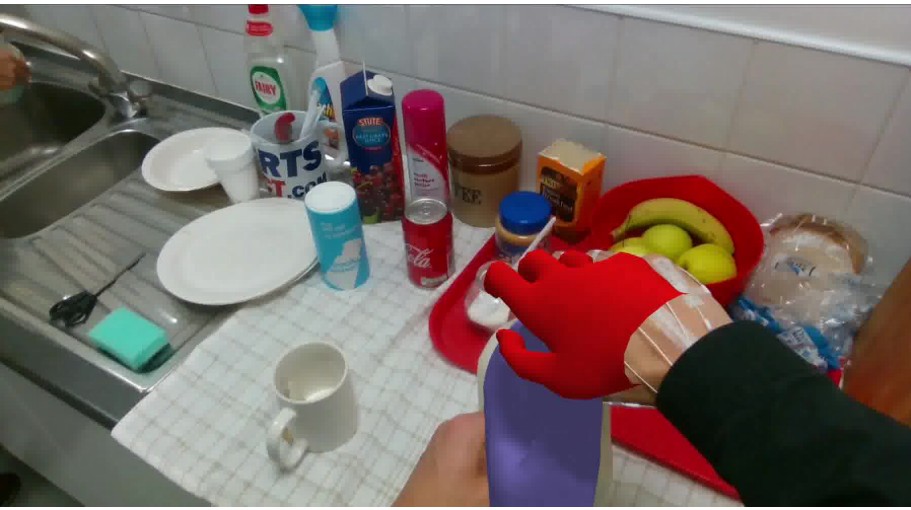

**Figure 1:** We forward the MANO model to convert the outputs of Inverse MANO Retargeting Network $\phi$ to hand meshes, which are projected to the original frames in FPHA with the help of camera intrinsics and extrinsics.

## A.3 Flow Matching

Recent work in high-resolution image and video synthesis has shown that flow matching can achieve strong empirical performance when combined with a simple linear-Gaussian (or optimal transport) probability path, given by:

$$q(\mathcal{H}_k^\tau | \mathcal{H}_k) = \mathcal{N}(\tau \mathcal{H}_k, (1-\tau)^2 \mathbf{I}).$$

In practice, the network is trained by sampling random noise $\epsilon \sim \mathcal{N}(0, \mathbf{I})$, computing the "noisy actions" $\mathcal{H}_k^\tau = \tau \mathcal{H}_k + (1-\tau)\epsilon$, and then training the network outputs $\nu_\theta(\mathcal{H}_k^\tau, h_k, z_k^{TDI})$ to match the denoising vector field:

$$\mathbf{u}(\mathcal{H}_k^\tau | \mathcal{H}_k) = \epsilon - \mathcal{H}_k.$$

The action expert uses a full bidirectional attention mask, so that all action tokens attend to each other. During training, we sample the flow matching timestep $\tau$ from a beta distribution that emphasizes lower (noisier) timesteps. At inference time, we generate actions by integrating the learned vector field from $\tau = 0$ to $\tau = 1$, starting with random noise $\mathcal{H}_k^0 \sim \mathcal{N}(0, \mathbf{I})$. We use the forward Euler integration rule:

$$\mathcal{H}_k^{\tau+\delta} = \mathcal{H}_k^\tau + \delta \nu_\theta(\mathcal{H}_k^\tau, h_k, z_k^{TDI}),$$

where $\delta$ is the integration step size. We use 10 integration steps (corresponding to $\delta = 0.1$) in our experiments. Note that inference can be implemented efficiently by caching the attention keys and values for the prefix $h_k, z_k^{TDI}$ and only recomputing the suffix corresponding to the hand motion for each integration step.

## A.4 Vision Language Model

Visual inputs are resized to $224 \times 224$ and encoded by SigLIP-2 with pixel shuffle [33], producing 64 spatially-aware visual tokens per frame, denoted as $x^\mathcal{I}$. In parallel, textual instructions are processed by SmolLM2 to extract semantic representations $x^\mathcal{T}$, facilitating cross-modal alignment.

## A.5 HOI Datasets

**Resources**. We utilize a variety of publicly available egocentric hand-object interaction datasets in our experiments. Below is a brief description of each dataset along with its official website for reference:

- **H2O**: A large-scale egocentric dataset featuring hand-object interactions with both RGB and depth modalities. https://taeinkwon.com/projects/h2o/
- **HOI4D**: A dataset of human-object interactions, capturing fine-grained manipulation across various tasks. https://hoi4d.github.io/
- **HOT3D**: A dataset for hand-object tracking and manipulation with accurate annotations. https://facebookresearch.github.io/hot3d/
- **OAKINK2**: A comprehensive benchmark for large-scale egocentric manipulation with articulated object models. https://oakink.net/v2/
- **TACA**: A task-oriented dataset for contact-aware human-object interaction analysis. https://taco2024.github.io/
- **ARCTIC**: A richly annotated dataset for tracking hand-object contact and motion in egocentric scenarios. https://arctic.is.tue.mpg.de/
- **HOLO**: A large-scale dataset of household manipulation tasks captured in real-world environments. https://holoassist.github.io/#HoloAssist

**Format**. Our training corpora are built upon the LeRobot [5] dataset format, a widely used standard in the open-source robotics community and interaction learning community. Developed by Hugging Face, LeRobot is designed to make it easier to work with demonstration-based learning by offering a unified structure for storing, sharing, and utilizing demonstration data. Its popularity stems from its adaptability and the rich ecosystem of pretrained models and datasets available on the Hugging Face hub. The LeRobot format combines several well-established file types to ensure efficient storage and accessibility:

- **Tabular Data:** States, actions, and metadata are stored in Parquet files, which provide compact columnar storage and rapid access. This structure supports fast filtering and slicing—critical for training modern machine learning models.

- **Visual Data:** Observations in the form of videos (MP4) or image sequences (PNG) are referenced in the Parquet files, significantly reducing storage requirements while preserving accessibility.

- **Metadata:** Supplementary information such as dataset statistics and episode indexing is stored in JSON format, allowing structured, machine-readable access to dataset characteristics.

Demonstration sequences are organized into episodes, where each frame captures synchronized observations and corresponding actions. Observations typically include visual inputs (e.g., `observation.images.*`) and internal states (e.g., `observation.state`), while actions encode control directives. This episodic structure supports a wide range of learning paradigms. For imitation learning, the data enables supervised prediction of actions from observations. For reinforcement learning, it facilitates evaluation and optimization of decision-making strategies under varied state-action contexts. This standardized data format not only enhances reproducibility and interoperability across learning systems but also lowers the barrier to entry for researchers by providing a clean interface to high-quality interaction datasets.

While the LeRobot format provides a solid foundation, our work introduces several extensions to accommodate richer modality integration. We augment the standard format with the following components:

- **Modality Configuration:** A `modality.json` file is introduced within the `meta` directory to explicitly define the structure of the initial state and action vectors. This configuration maps each vector component to its semantic meaning and includes additional metadata relevant to each modality.

- **Fine-Grained Semantic Decomposition:** Departing from the monolithic vector approach of the original format, we decompose both state (initial hand state) and action (future hand motion trunk) vectors into semantically interpretable components—such as $\theta$, $\beta$, $r$, and $t$—each annotated with its own data type, valid range, and transformation rules.

- **Multi-Annotation Integration:** The dataset format is extended to support multiple forms of annotations, such as task descriptions, validity indicators, and success labels. These annotations follow the LeRobot practice of storing indices in the Parquet files, with the corresponding content stored in auxiliary JSON files.

- **Rotation Representation Specification:** To ensure correct processing of rotational components during training, we require explicit declaration of the rotation representation used (e.g., quaternion, Euler angles, or axis-angle) for each relevant field.

These enhancements collectively enable more structured learning from complex demonstration data, with explicit modality definitions and robust support for multimodal supervision.

**Preprocessing**. For FPHA, we pretrain Inverse MANO Retargeting Network to label MANO parameters. For ARCTIC, HOT3D and OAKINK2, we adopt virtual RGB-D rendering to produce high-quality metric depth images in advance. All RGB and depth images are resized to $256 \times 256$. It is worth noting that we split longer sequences to short clips (<500 steps) with the same task instruction for training and testing.

### A.6 Computation

**Resources**. MEgoHand is trained using 8×80GB NVIDIA A800 GPUs over approximately 24 hours. All evaluations and visualizations are performed on a single 80GB A800 GPU for around three hours.

**Efficiency**. We evaluated the end-to-end inference performance for generating a 16-frame sequence on a single RTX 4090 GPU. MEgoHand is over 2x faster and uses nearly 50% less VRAM than the strong LatentAct baseline. This superior efficiency is a direct result of our novel architectural design, which eliminates the need for the expensive online contact map computation. Instead, MEgoHand leverages a lightweight, pre-trained depth estimation module that implicitly captures geometric cues

while operating at a fraction of the computational cost. This design not only reduces inference latency and memory footprint but also enhances robustness across diverse interaction scenarios, making it well-suited for real-time AR/VR and robotic applications.

**Table 2:** End-to-end inference performance efficiency for generating a 16-frame sequence on a single RTX 4090 GPU.

| Method | Inference Time↓ | FPS↑ | VRAM Usage↓ |
|---|---|---|---|
| LatentAct | 156ms | 6.4 | 10.8GB. |
| **MEgoHand (Ours)** | **74ms** | **13.5** | **5.8GB** |

## A.7 Smooth Decoding

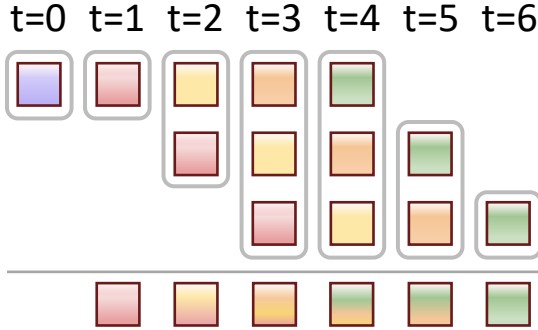

**Figure 2:** Illustration for smoothing predicted transformations.

**Decoding Strategy**. As illustrated in Figure 2, at $t = 0$ MEgoHand receives initial hand MANO parameters, a egocentric RGB observation, and a depth map to predict trunk $t = 1 \cdots l$. The predicted wrist pose is relative to the initial hand pose and the predicted $\hat{\beta}$ is repeated from initial $\beta$. Then at $t = 1$, similarly, the predicted wrist pose $t = 2 \cdots l + 1$ is relative to the wrist pose predicted at $t = 1$, and so on. After converting all relative transformations to absolute transformations, we average all predictions at the same timestep to get smoother transformations.

**Visualization**. From Figure 3 we can see that smooth decoding stategy is effective in mitigating jitter.

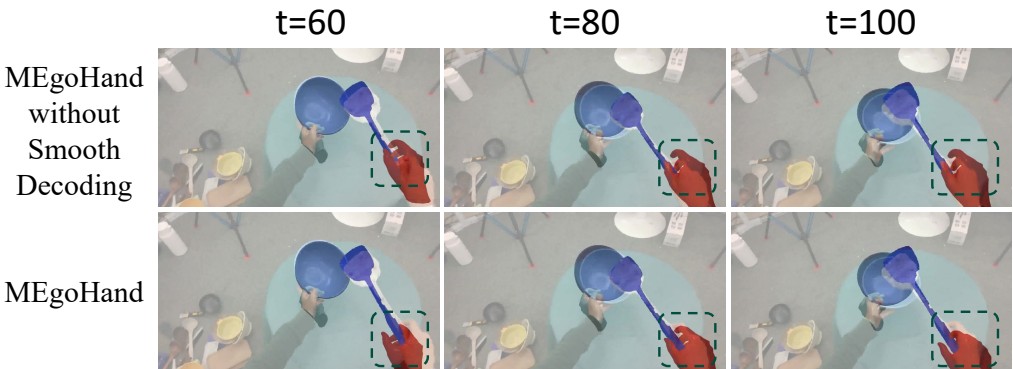

**Figure 3:** Frames randomly sampled from task *"Stir the bowl with spatula"* of TACO. Without decoding strategy, the predicted trajectory exhibits more fluctuations.

# B Additional Visualizations

## B.1 Zero-Shot Depth Estimation & Virtual Depth Rendering

In Figure 4, we visualize the zero-shot depth estimation of UniDepthV2 [29] and the virtual depth rendered from object models. Three datasets (OAKINK2,HOT3D,ARCTIC) are involved, as there are no real depth frames in these datasets.

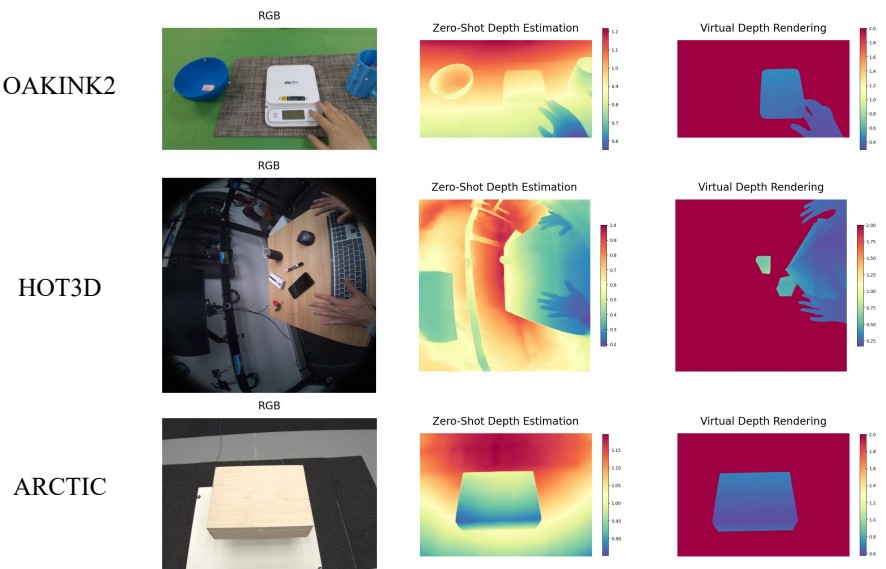

**Figure 4:** Colorbars indicate the absolute depth values (unit: m). The depth values of all depth frames fall within $[0, 2]$.

## B.2 HOI hand motion Generation

We visualize more clips of policy inference in Figures 6 and 5. MEgoHand is superior to baseline LatentAct in most cases.

# C Empirical Results

We report the average metrics of MEgoHand in each dataset in Table 3.

**Table 3:** Average metrics across evaluation (TACO, HOI4D, H2O, HOT3D, OakInk2) and testing datasets (ARCTIC, HOLO). The unit for MRE is radians; the remaining metrics are measured in centimeters.

| Dataset | MPJPE | MPJPE-PA | MPVE | MPVE-PA | MWTE | MRE |
|---------|-------|----------|------|---------|------|-----|
| H2O     | 3.013 | 0.352    | 2.969 | 0.334   | 2.450 | 0.099 |
| HOI4D   | 8.958 | 0.856    | 8.933 | 0.826   | 8.462 | 0.213 |
| HOT3D   | 6.437 | 0.236    | 6.352 | 0.228   | 5.045 | 0.086 |
| OAKINK2 | 3.424 | 0.217    | 3.380 | 0.205   | 2.837 | 0.071 |
| TACO    | 4.936 | 0.358    | 4.899 | 0.346   | 4.465 | 0.131 |
| ARCTIC  | 7.358 | 1.161    | 7.268 | 1.106   | 5.958 | 0.398 |
| HOLO    | 5.775 | 0.697    | 5.747 | 0.673   | 5.437 | 0.271 |

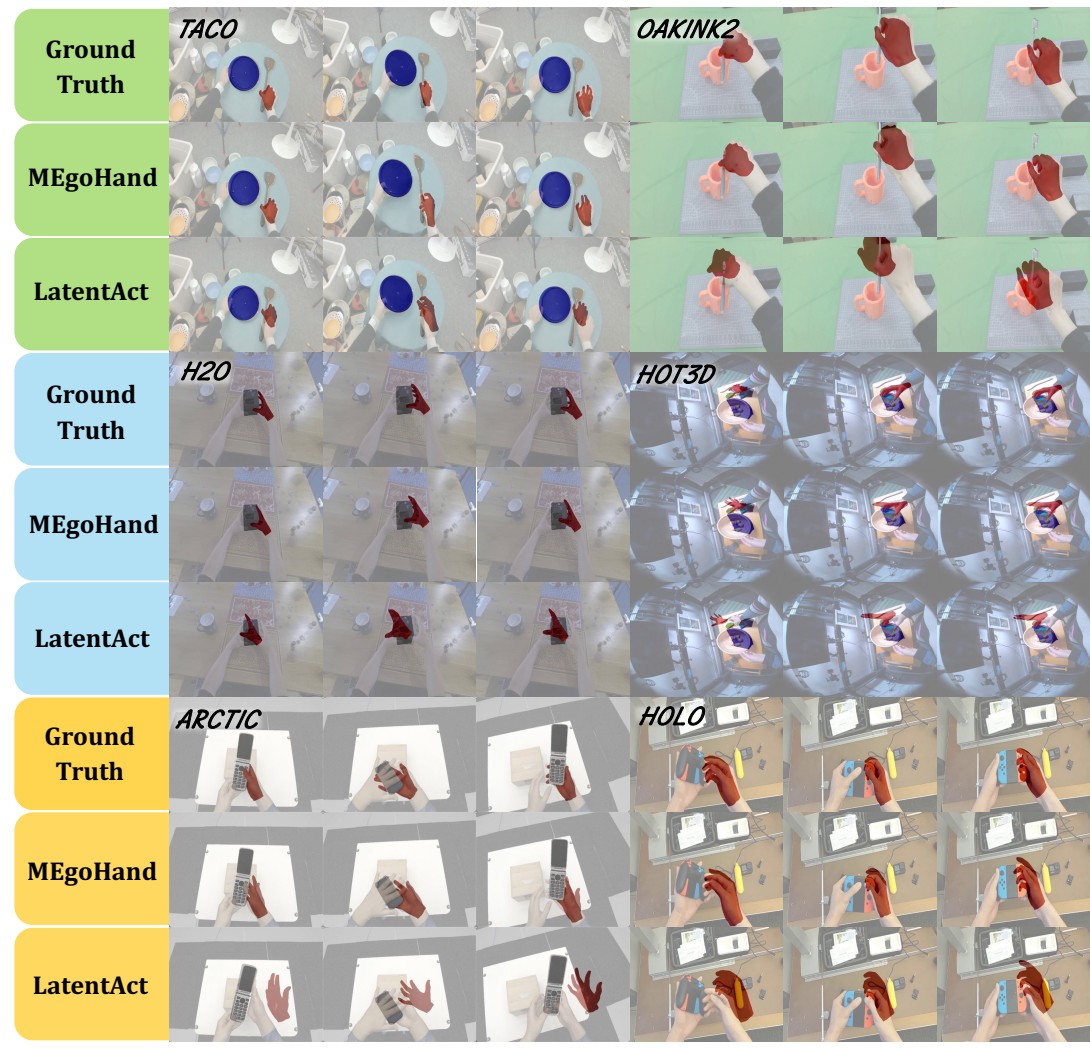

**Figure 5:** Additional visualizations of LatentAct and MEgoHand. Green part is sampled from training sets. Blue part is sampled from evaluation sets. The Yellow part is sampled from testing sets.

# D Social Impact

MEgoHand forwards an important step toward universal hand-object motion generation from multiple modalities including task instruction, RGB observation, depth image, and initial conditions. There are many potential societal consequences of our work, none which we feel must be specifically highlighted here.

