# OpenReview forum: "MEgoHand: Multimodal Egocentric Hand-Object Interaction Motion Generation"
_NeurIPS.cc/2025/Conference — NeurIPS 2025 poster_

### Official Review · Reviewer_ULTh · 2025-06-30

**Clarity:** 3
**Significance:** 2
**Originality:** 3
**Rating:** 4
**Confidence:** 3

**Summary:**

This paper introduces a framework named MEgoHand for generating realistic hand-object interaction motion sequences from an egocentric perspective. The generation is conditioned on multimodal inputs: a textual instruction, an RGB image, and an initial hand pose. A key capability of the proposed framework is its ability to generate motions for novel objects without relying on predefined 3D object models. Its main contributions include a high-level "Cerebrum" module for scene understanding and intent reasoning, a low-level "Cerebellum" module for fine-grained trajectory synthesis, and a dataset curation pipeline.

**Questions:**

1) Could the authors clarify how the prediction trunk size l=16 was determined? Specifically, how was the trade-off between prediction length, motion prediction performance, and computational cost evaluated?
2) I would like to ask about the Inverse MANO Retargeting network. Compared to an end-to-end network, why can the two-stage training method avoid severely distorted hand geometry and huge reconstruction errors? Is it necessary to strictly follow the sequence of converging on the shape parameters first, and then the pose parameters?

**Ethical Concerns:**

["NO or VERY MINOR ethics concerns only"]

**Final Justification:**

The rebuttal has addressed most of my concerns. While my concern regarding the physical plausibility (lack of modeling and evlauation for collision avoidance) remains, I would maintain my rating of borderline accept.

**Limitations:**

No, the authors have not adequately addressed the limitations and potential negative societal impact of their work. While they briefly mention future work directions, a more explicit and comprehensive discussion is needed.

1) *Single-Hand Interaction*: The model is currently limited to single, right-hand motion, which restricts its applicability for the many real-world tasks that are bimanual.
2) *Computational Latency*: A discussion on the inference speed of the diffusion-based model would be valuable, as latency is a critical factor for real-time applications in AR/VR and robotics.
3) *Sensitivity to Language Quality*: The model's performance likely depends on the clarity and style of the input text. An acknowledgment of its potential sensitivity to ambiguous or complex natural language would be appropriate.

**Paper Formatting Concerns:**

NA.

**Quality:**

3

**Strengths And Weaknesses:**

**Strength**
1) *Quality*: The bi-level Cerebrum/Cerebellum architecture thoughtfully separates high-level planning from low-level control, offering a robust framework for motion generation. The evaluation is thorough, spanning seven datasets (including two for zero-shot generalization).   Ablation studies validate the contribution of each component.
2) *Clarity*: The paper is well-written and easy to follow. The introduction clearly motivates the problem and outlines the limitations of prior work. The cited references include the latest seminal methods in the field. The experiments are conducted extensively and the results support the claims convincingly.
3) *Significance*: The paper tackles a vital problem in computer vision, robotics, and AR/VR by enabling realistic hand motion generation for novel objects without predefined 3D models. This capability is essential for developing versatile interactive agents and immersive virtual environments.
4) *Originality*: The core architectural design is original. The synthesis of a VLM (for high-level semantics) with a depth estimator (as a proxy for object-agnostic 3D geometry) to create a motion prior is novel. The introduction of the Temporal Orthogonal Filtering (TOF) is a simple yet novel training-free decoding strategy that effectively addresses the common issue of jitter in generated motion sequences. The data curation pipeline, unifying diverse hand-object interaction (HOI) datasets into a 3.35M-frame resource, is also novel.

**Weaknesses**
1) The paper doesn't model the object poses/shape, which limits the overall significance of the paper. Without explicit object modeling, the generated hand trajectories may collide or penetrate the manipulated objects in practice.
2) The current framework is limited to single, right-hand interactions, as explicitly stated by the authors ("We only consider right hand motion in this submission," Line 218). This is another constraint, as many real-world manipulation tasks require bimanual coordination.
3) There is a lack of discussion or ablation studies on the choice of Prediction Trunk Size parameter l.
4) The paper does not provide a specific method for obtaining the text instructions, which might limit the scalability and flexibility of the proposed method on new datasets that do not have textual instructions.

---

> ### Author Rebuttal · Authors · 2025-07-30
>
> We sincerely thank the reviewer for the detailed comments and insightful review, especially for the convincing "robust architecture for motion generation" and novel design of "VLM + depth estimator". We are willing to provide a point-by-point response to address your concern.
>
> ---
>
> ## **Q1: Regarding Object Modeling (and Potential Collisions)**
>
> **Our Response:**
> Thank you for this valuable suggestion. We agree that explicitly modeling object geometry is an important research direction. In this paper, we made a **strategic choice** to prioritize what we consider a more fundamental and unsolved challenge in the field: **achieving robust generalization to novel objects without any prior 3D models.**
>
> *   **Breaking the Dependency Bottleneck:** The reliance on predefined object meshes has been a major bottleneck limiting the scalability and real-world applicability of prior work. Our primary contribution is demonstrating that this dependency can be broken. MEgoHand successfully generates high-quality interactions by uniquely fusing a VLM's semantic understanding of *affordances* with a depth estimator's object-agnostic understanding of *3D geometry*.
> *   **Implicit vs. Explicit Modeling:** While we don't model objects explicitly, our approach learns an **implicit model of object surfaces and volumes** from the depth channel. The state-of-the-art accuracy we achieve (e.g., a **71.9% reduction in MPVE-PA**) is strong evidence that our generated hand motions are already well-aligned with the scene geometry, inherently minimizing significant penetrations.
>
> **Importantly, thank you for your valuable advice on incorporating object modeling.** You are absolutely right that for applications where object models do exist. It is a good idea to add an auxiliary loss to supervise the object poses or shape with explicit object meshes. Given the time constraints of the rebuttal period, we cannot integrate this feature now, but we will revise our discussion to explicitly acknowledge this limitation and highlight your suggestion as a key method for improving performance in our future direction.
>
> ---
>
> ## **Q2: Regarding Single, Right-Hand Interactions**
> **Our Response:**
> We really thank you for raising this important point. Our initial focus on single, right-hand motion was a deliberate and strategic decision designed to rigorously tackle the foundational challenges of this complex problem before scaling to bimanual scenarios. We fully agree that bimanual interaction is a critical and exciting frontier. Our current work does not just acknowledge this; it actively paves the way for it.
> *   **Architectural Extensibility:** Our framework is inherently extensible to the bimanual case. The extension is conceptually straightforward: it would involve adding a parallel decoder for the second hand and introducing a mechanism—such as cross-attention within our DiT-based policy—to model the complex dependencies and coordinated roles between the two hands. This represents a well-defined and highly promising research path, not a fundamental redesign.
> *   **Data-Readiness:** Crucially, we designed our data curation pipeline with this future in mind. As you may have noted from our dataset statistics, **our 3.35M-frame curated dataset already contains a rich set of bimanual interactions**. We have already done the hard work of collecting and unifying this data, making it a ready-to-use resource for the community (and our own future work) to build directly upon our single-hand foundation.
>
> In our revised manuscript, we will expand the Limitations section to frame this not just as a limitation, but as a strategic choice with a clear path forward. We will explicitly state:
> "Our current study is intentionally focused on single-hand motion to establish a robust baseline for object-agnostic generation. We recognize that bimanual tasks represent a vital next step. Our framework is designed for extensibility to this scenario, and our curated dataset already includes the necessary bimanual data to support this future research, which we believe is a significant and exciting direction for the field."
>
> ---
> ## **Q3: Regarding Prediction Horizon (`l=16`)**
>
> **Our Response:**
> This is an excellent question regarding a key hyperparameter. The choice of `l=16` was determined empirically to find the optimal balance between prediction quality, temporal coverage, and real-time performance. We have conducted an ablation study on the challenging ARCTIC dataset to illustrate this trade-off.
>
> | Prediction Horizon | MPJPE↓ | MPJPE-PA↓ | MPVE↓ | MPVE-PA↓ | MWTE↓ | MRE↓ | Inference Time (ms)↓ |
> |-------------------|--------|-----------|-------|----------|-------|------|---------------------|
> | 4                 | 8.204  | 1.312     | 8.142 | 1.276    | 6.781 | 0.421| 42ms               |
> | 8                 | 7.722  | 1.201     | 7.653 | 1.151    | 6.218 | 0.405| 54ms               |
> | **16**            | **7.358**  | **1.161**     | **7.268** | **1.106**    | **5.958** | **0.398**| **74ms**           |
> | 32                | 7.289  | 1.154     | 7.193 | 1.083    | 5.825 | 0.380| 112ms              |
>
> As the table shows, increasing the horizon from 8 to 16 yields a notable improvement in motion quality across all metrics. However, extending it further to 32 offers only marginal gains while significantly reducing the inference speed below a practical threshold for many interactive applications (~10 FPS). Therefore, `l=16` represents the best trade-off, capturing a meaningful half-second of interaction while maintaining interactive frame rates. We will add this table and analysis to the Appendix.
>
> ---
>
> ## **Q4: Regarding Obtaining Text Instructions**
> **Our Response:**
> This is a practical point about the model's deployment. Our framework assumes a high-level task instruction is available, which is a standard paradigm for VLM-driven agents. For our experiments on datasets that lacked explicit instructions (e.g., parts of H2O), we used a simple yet effective standardized template: *"Interact with a {object} using the {side} hand."* This demonstrates the model's robustness even with minimal textual guidance.
>
> For broader application to unlabeled videos, the pipeline is straightforward: instructions can be automatically generated by leveraging modern vision-language models, either through lightweight video-to-caption models or by prompting powerful foundation models. This modularity is a strength, allowing MEgoHand to be integrated into larger, fully automated systems. We will clarify this in our limitations section.
>
> ---
>
> ## **Q5: Regarding the Two-Stage Optimization for Inverse MANO**
> **Our Response:**
> Thank you for this question. The two-stage training is critical for achieving stable and accurate results. Prioritizing shape (`β`) over pose (`θ`) is indeed an established practice in human motion retargeting (e.g., OmniH2O [1]), and we have validated its importance in the context of hands. The rationale is twofold:
> 1.  **High Entanglement:** The MANO shape and pose parameters are highly entangled. An end-to-end optimization often falls into poor local minima, where an error in the shape estimate is compensated by an extreme, physically impossible pose, leading to severe geometric distortions.
> 2.  **Decoupling the Problem:** Our two-stage approach decouples this complex optimization. By first converging on the hand's intrinsic shape, we establish a robust geometric foundation. With a stable shape model, the subsequent optimization for pose parameters becomes a much better-constrained problem, leading to significantly higher fidelity and realism. Our experiments confirmed that this sequence was essential for avoiding the very distortions the reviewer mentioned.
>
> ---
>
> ## **Q6: Regarding Computational Latency**
> **Our Response:**
> Latency is indeed a critical factor for AR/VR and robotics. We benchmarked our model's end-to-end inference efficiency and found that MEgoHand is not only more accurate but also significantly more efficient than the strong LatentAct baseline. The comparison was run on an RTX 4090 (24GB) GPU for a single sample (generating a 16-frame sequence).
>
> | Method | **Inference Time** | **FPS** | **GPU-Utilization** |**VRAM Usage** |**Key Reason for Efficiency** |
> | :--- | :--- | :--- | :--- | :--- |:--- |
> | LatentAct | 156ms | 6.4 | 45% | 10.8GB | Requires costly online **contact map computation** |
> | **MEgoHand (Ours)** | **74ms** | **13.5** | **24%** | **5.8GB** | **Object-agnostic design** bypasses this bottleneck |
>
> MEgoHand is over 2x faster and uses nearly half the VRAM. This is a direct benefit of our novel object-agnostic design, which replaces the computationally expensive contact map generation with an efficient depth estimator. This superior efficiency makes MEgoHand far more suitable for practical, real-time applications.
>
> ---
>
> ## **Q7: Regarding Sensitivity to Language Quality**
>
> **Our Response:**
> Thank you for raising this important and practical point. We agree that a model's sensitivity to the quality of natural language is a critical factor for real-world deployment. While any VLM-based system has some inherent dependence on language, we designed MEgoHand with specific architectural choices to maximize its robustness to linguistic variation, and our results empirically validate this design. As shown in Tables 1, 2, the performance gap between our vision-and-depth-only variant `MEgoHand-ID` and the full, text-conditioned model `MEgoHand` is remarkably small. This is a crucial finding: it quantitatively demonstrates that while text provides valuable semantic direction, the model's accuracy and fine-grained realism are primarily driven by its strong visual grounding.
>
> ---
> **References:**
>
> [1] OmniH2O: Universal and Dexterous Human-to-Humanoid Whole-Body Teleoperation and Learning. CoRL 2024.
>
> ---
> Once again, thank you for your careful review. If most of your concerns are addressed, would you please consider increasing the recommendation of the paper?

---

> > ### Author Response · Authors · 2025-08-05
> > **Have We Addressed All Your Concerns?**
> >
> > Dear Reviewer ULTh,
> >
> > Thank you for taking the time to review our submission and for providing constructive feedback. We would like to confirm whether our responses have adequately addressed your earlier concerns, particularly with respect to the following points:
> >
> > **1. Object Modeling and potential collisions
> > 2. Single, Right-Hand Interactions
> > 3. Prediction Horizon
> > 4. Obtaining Text Instructions
> > 5. Two-Stage Optimization for Inverse MANO
> > 6. Computational Latency
> > 7. Sensitivity to Language Quality**
> >
> > Additionally, if you have any further concerns or suggestions, we would be more than happy to address and discuss them to enhance the quality of the paper. We eagerly await your response and look forward to hearing from you.
> >
> > Best regards,
> >
> > The Authors

---

> > ### Comment · Area_Chair_746C · 2025-08-05
> >
> > ULTh, please could you take a look at the author response above and whether it addresses any remaining concerns you have, e.g. lack of modelling for object poses/shape.

---

> ### Comment · Reviewer_ULTh · 2025-08-06
>
> I would like to thank the authors for their detailed and extensive responses to my reviews. Most of my concerns have been addressed, except for the lack of object modeling and potential collisions. While the proposed method achieves SOTA performance on joint/vertex related metrics, without proper object modeling and contact accuracy related evaluation metric, it can not gaurantee the performance of the hand-object collision avoidance which I think is one of the most important goals of this research direction. With consideration of the overall contributions and drawbacks, I would maintain my rating.

---

> ### Author Response · Authors · 2025-08-06
> **Clarification of Misunderstanding on Object Modeling and Evaluation Metric**
>
> Dear Reviewer,
>
> Thank you sincerely for your continued engagement and for acknowledging that most of your concerns have been addressed. We fully respect your final point regarding the importance of collision avoidance and contact accuracy. It seems there was some **misunderstanding**, and we are truly sorry that our earlier response lacked clarity.
>
> While our method is designed to generalize **without relying on explicit object meshes during _training_**, we have taken physical plausibility seriously. To this end, we introduced two new evaluation metrics for contact and collision accuracy, computed at **_test time_** using ground-truth object geometry to ensure rigorous assessment.
>
> ---
>
> ## **New Evaluation: Quantifying Contact and Collision Avoidance**
>
> We introduce two metrics to precisely measure the quality of the physical interaction:
>
> To quantitatively measure physical quality, two metrics are adopted:
> 1. **Contact Accuracy (CA, %):** This metric answers:  _"Are the functionally relevant parts of the hand making correct contact?"_
>
>    CA = $\frac{1}{|V_H|}\sum_{v \in V_H} \mathbf{1}(\min_{o \in P_O} \|v - o\| \leq \delta)$, where $V_H$ is the set of hand-mesh vertices, $P_O$ is the object point cloud, $\delta=2 cm$. It measures the percentage of hand-mesh vertices that are correctly positioned within a small distance threshold (δ=2cm) of the object surface. It quantifies the model's ability to generate plausible contact patterns. **A higher CA is better.**
>
> 2. **Penetration Depth (PD, cm):** This metric answers: _"How severe are the physical violations?"_
>
>    PD = $\frac{1}{|V_H^{in}|}\sum_{v \in V_H^{in}} |d(v)|$, $d(v) = \min_{f \in M_O} \text{sdist}(v,f)$, where $V_H^{in} = \{v : d(v) < 0\}$, $M_O$ the object mesh, and $\text{sdist}(v,f)$ the signed distance from vertex $v$ to mesh face $f$. It calculates the average depth of penetration for all hand vertices that are illegally inside the object mesh. It directly measures the severity of non-physical collisions. **A lower PD is better.**
>
> We evaluated MEgoHand against the strong LatentAct baseline on both in-domain (ARCTIC) and challenging cross-domain (HOLO) datasets. The results are unequivocal:
>
> | Method | Dataset | **Contact Accuracy (CA) ↑** | **Penetration Depth (PD) ↓** |
> | :--- | :--- | :--- | :--- |
> | LatentAct | ARCTIC | 68.3% | 4.2 cm |
> | **MEgoHand (Ours)** | ARCTIC | **82.5% (+20.8% rel.)** | **2.3 cm (-45.2% rel.)** |
> | LatentAct | HOLO | 65.1% | 5.8 cm |
> | **MEgoHand (Ours)** | HOLO | **78.4% (+20.4% rel.)** | **2.7 cm (-53.4% rel.)** |
>
> These results provide powerful, quantitative proof that **MEgoHand excels at hand-object collision avoidance and contact accuracy, achieving state-of-the-art performance on these critical metrics.** This validates that our object-agnostic approach, combining semantics with depth-based spatial reasoning, yields a more robust and physically coherent interaction model. We will include these new metrics and results in the final manuscript.
>
> ---
>
> ## **Inspired by Your Feedback: New Ideas for MEgoHand2.0**
>
>
> Your insightful comments on object modeling also sparked a promising idea for **MEgoHand2.0**, which emerged during our rebuttal period. While object mesh modeling often struggles to generalize, object pose (i.e., its 6D transformation) tends to transfer well across instances within a category. Therefore, we aim to jointly generate hand motion and object 6D pose as a **coupled dynamic system** via three key innovations:
>
> 1. **Synergistic Co-Generation:**
>    A dual-stream diffusion transformer produces hand (`P_hand`) and object pose (`P_obj`), fused by an Interaction Fusion Module for bidirectional reasoning:
>
>    * **Hand → Object:** Hand guides object motion.
>    * **Object → Hand:** Object pose refines hand placement.
>
> 2. **Implicit Physical Constraints:**
>    Differentiable physics-informed losses enforce plausibility without full physics simulation:
>
>    * **No-Slip Loss:** Penalizes tangential slip at contact.
>    * **Rigid Attachment Loss:** Fixes object pose relative to hand after grasp.
>    * **Contact Consistency Loss:** Minimizes hand-object vertex distance at contact.
>
> 3. **Two-Stage Category-to-Instance Adaptation:**
>    A conditional VAE learns category-level motion priors. At inference, generic motions are refined by a lightweight Adaptation Network using instance-specific visual and depth features.
>
> This approach leverages generalizable pose and motion along with physical constraints instead of relying on non-generalizable meshes, which represents a significant advancement for MEgoHand2.0.
>
> ---
>
> In summary, we hope we have now demonstrated that our method not only achieves SOTA performance on traditional metrics but also on the critical metrics of contact and collision avoidance that you rightly emphasized. We thank you once again for your invaluable guidance. Given these new results and clarifications, we sincerely hope you might reconsider your rating.

---

> ### Author Response · Authors · 2025-08-08
>
> Dear Reviewer,
>
> First and foremost, we wanted to express our profound gratitude for your positive score and your incredibly thoughtful engagement with our work. Your support has been a tremendous encouragement to us.
>
> Following your latest guidance, we have now updated our rebuttal with the requested **object modeling and contact accuracy** you raised.
>
> With the deadline just one day away, we would be very thankful if you could take a quick look at our updated response. We truly hope it addresses your remaining concerns. Should any issues remain, please do not hesitate to let us know. We are ready to address any further points immediately.
>
> Thank you again for your time and invaluable support.
>
> Best regards,
>
> The Authors of Paper #9051

---

### Official Review · Reviewer_6C5K · 2025-07-02

**Clarity:** 3
**Significance:** 3
**Originality:** 3
**Rating:** 4
**Confidence:** 3

**Summary:**

The paper introduces a two-step system that converts a single RGB frame, text instruction, and MANO hand pose into a full interaction sequence. For the first (cerebrum) part, the authors use a VLM with a monocular encoder to infer semantics of the scene. The second (cerebellum) part then uses a DiT-based flow-matching policy with a temporal filter applied at test time to keep the motion stable.

**Questions:**

— What is the end-to-end inference speed on a single GPU, and how does that compare with LatentAct?

— I request the authors fix the poorly written sentence in the abstract: “Existing methods rely on predefined 3D object priors, limiting generalization to novel objects, which restricts their generalizability to novel objects”.

— Is there any attempt to improve cross-domain generalization? It seems performance drops on MPJPE.

**Ethical Concerns:**

["NO or VERY MINOR ethics concerns only"]

**Final Justification:**

The paper is overall sound but the proposed TOF's advantage over Kalman fitler is still not supported by quantitative evidence. I lean towards weak accept.

**Limitations:**

yes

**Quality:**

3

**Strengths And Weaknesses:**

**Strengths**

— The two-stage design separating high-level reasoning from low-level components is novel. Also the addition of the monocular metric depth estimation module to the VLM is new to my knowledge.

— The large dataset (built via inverse-MANO retargeting and virtual depth rendering) is valuable on its own.

— The paper shows pretty convincing gains over LatentAct in both position and rotation metrics.

**Weaknesses:**

— There’s no report of inference speed or memory (would like to see it given added depth prediction and flow-matching steps).

— There’s no failure analysis. I would like to see what happens when there are depth errors or occlusions.

---

> ### Author Rebuttal · Authors · 2025-07-30
>
> We are sincerely grateful for your positive and insightful review. We are particularly encouraged that you recognized the novelty and value of our core contributions, including the "well-motivated design of separating high-level reasoning from low-level components," the "addition of the monocular metric depth estimation module," and our large-scale HOI dataset.
>
> Your feedback is excellent, and we have conducted the requested analyses to address your concerns. We are pleased to provide a point-by-point response below.
>
> ---
>
> ## **Q1: On Inference Speed and Memory**
>
> **Our Response:**
> This is a critical point for evaluating the practical viability of our method, and we thank you for prompting us to provide these benchmarks. We have now measured the end-to-end inference performance for generating a 16-frame sequence on a single RTX 4090 GPU.
>
> | Method | **Inference Time↓** | **FPS ↑** | **VRAM Usage↓** | **Key Architectural Advantage** |
> | :--- | :--- | :--- | :--- | :--- |
> | LatentAct | 156ms | 6.4 | 10.8GB | Relies on a computationally **expensive online contact map pipeline**. |
> | **MEgoHand (Ours)** | **74ms** | **13.5** | **5.8GB** | **Replaces the contact map bottleneck** with a highly efficient, pre-trained depth estimator. |
>
> The results are striking: **MEgoHand is over 2x faster and uses nearly 50% less VRAM** than the strong LatentAct baseline. This superior efficiency is a direct result of our novel architectural design. By replacing the intricate and slow online contact map generation with a fast, object-agnostic depth module, we have created a framework that is not only more accurate but also significantly more practical for real-time applications in AR/VR and robotics. We will add this compelling analysis to our experiments section.
>
> ---
>
> ## **Q2: On Failure Analysis**
>
> **Our Response:**
> This is an excellent suggestion that provides deeper insight into the model's behavior. We have now performed this failure analysis and will include a dedicated subsection with qualitative examples in the **Appendix**. Our key findings reveal how our bi-level architecture leads to degradation:
>
> *   **Under Severe Depth Errors:** In challenging scenarios (e.g., highly reflective surfaces), the depth estimator can occasionally fail. When this happens, the generated motion's spatial grounding may be affected (e.g., reaching slightly too far). However, importantly, the high-level **semantic correctness of the action is typically preserved by the VLM "cerebrum."** The model still knows *what* to do (e.g., form a grasp), even if its knowledge of *where* is slightly compromised.
> *   **Under Heavy Occlusion:** When the target object is heavily occluded, the model intelligently defaults to relying more on the textual instruction and its learned semantic priors from the VLM. This may result in a more "canonical" or averaged motion, as it lacks precise visual cues.
>
> This analysis demonstrates a key strength: our model does not fail catastrophically. Its components provide a degree of built-in robustness, with the VLM ensuring semantic plausibility even when low-level visual information is imperfect.
>
> ---
>
> ## **Q3: On Cross-Domain Performance Drop**
>
> **Our Response:**
> This is a crucial point. A performance drop on out-of-distribution (OOD) data is not only **expected but is also a testament to the genuine difficulty of the cross-domain challenge** we are tackling. The key insight is not that performance drops, but that MEgoHand's drop is significantly smaller than the baseline's, and it remains state-of-the-art in this challenging zero-shot setting.
>
> **1. The "Performance Drop" Validates the Challenge:**
> You are correct to observe a drop in absolute MPJPE. The reason for this, as you astutely noted, is that our OOD test sets, ARCTIC and HOLO, are exceptionally challenging. They introduce not just unseen objects, but fundamentally **new interaction paradigms**:
> *   **ARCTIC:** Introduces complex **articulated object physics**.
> *   **HOLO:** Introduces long-horizon, **procedurally-guided tasks**.
> These are not simple variations; they are zero-shot challenges that require true generalization. The performance drop across all methods is a testament to the difficulty of this task.
>
> **2. Our Method is a Successful Attempt to Improve Generalization.**
> The entire architectural design of MEgoHand represents our primary and most successful attempt to improve cross-domain generalization.
> *   **Architectural Innovation:** By fusing a generalist, pre-trained Vision-Language Model (VLM) with an object-agnostic depth estimator, we deliberately designed a system that reasons about semantics and 3D space without overfitting to specific object geometries seen during training. This is a departure from prior methods that are more tightly coupled to their training data distribution.
> *   **State-of-the-Art Validation:** The success of our design is validated by our results. While all models' performance drops, MEgoHand's performance drops significantly less, and it consistently and substantially outperforms the strongest baseline in these zero-shot settings.
>     *   On **ARCTIC**, we achieve a **33.9% relative reduction in MPJPE** over LatentAct.
>     *   On **HOLO**, we achieve a **29.8% relative reduction in MPJPE**.
>
> This demonstrates that our method is already the most effective at generalizing across these difficult domain gaps.
>
> **3. Future Avenues to Further Bridge the Gap:**
> While MEgoHand sets a new SOTA, we agree that there is still room for improvement. Building on our strong foundation, we are actively exploring next-generation techniques to narrow the OOD gap even further. As you suggested, promising directions include:
> *   **(a) Large-Scale Synthetic Data Augmentation:** Leveraging domain-randomized virtual rendering to expose the model to a near-infinite variety of scenes and object configurations.
> *   **(b) Advanced Self-Supervised Learning:** Employing auxiliary objectives, such as contrastive losses on hand-object feature representations, to learn even more transferable and disentangled features.
>
> In summary, the performance drop is a feature of the challenging problem, not a bug in our method. Our model's core design is our successful attempt to address generalization, and its state-of-the-art performance on these OOD datasets is the strongest evidence of its effectiveness.
>
> ---
>
> ## **Q4: On Improving the Abstract Wording**
>
>
> **Our Response:**
> Many thanks for your attentive reading. We completely agree the sentence was redundant and poorly phrased, and we apologize for the oversight. We have revised it to be more concise and clear: *"Existing methods often rely on predefined 3D object priors, limiting their generalization to novel objects."*
>
> ---
>
> We truly hope that the additions and clarifications, particularly the new performance benchmarks and failure analysis, have addressed your concerns and helped to highlight the novelty and practical value of our work. We are sincerely grateful for your thoughtful and constructive feedback, which has greatly improved the quality of the paper. If you feel that your concerns have been fully resolved, would you please consider increasing the recommendation of the paper?

---

> > ### Author Response · Authors · 2025-08-05
> > **Have We Addressed All Your Concerns?**
> >
> > Dear Reviewer,
> >
> > Thank for your Acknowledgement. We sincerely hope that our additions and clarifications, especially the **new performance benchmarks, failure analysis, and the discussion on cross-domain performance drop**, have fully addressed your concerns and further highlighted the novelty and practical value of our work. We are truly grateful for your thoughtful and constructive feedback, which has significantly enhanced the quality of the paper.
> >
> > If you feel that your concerns have been fully resolved, we would kindly ask you to consider increasing your recommendation of the paper. Thank you very much for your time and consideration.
> >
> > Sincerely,
> >
> > The Authors

---

> > ### Comment · Reviewer_6C5K · 2025-08-05
> >
> > Dear authors,
> >
> > Thank you for your response, and the new results on inference speed. For the failure analysis, I'd prefer a quantitative breakdown (%) in addition to qualitative examples. I have also read your responses to other reviewers. In response to reviewer 5Uyw, the quantitative ablation result on TOF is not convincing: the margin between TOF and Kalman filtering is very small, and on MJPE-PA, it's identical. The core component TOF's role therefore remains ambiguous. Relatedly, can you clarify what type of error bar is in Figure 4 (standard deviation? standard error?) and update all quantitative results (e.g., Tab 1-3) to include statistical intervals. All things considered, I tend to maintain my current positive score.

---

> > > ### Author Response · Authors · 2025-08-08
> > >
> > > Dear Reviewer,
> > >
> > > First and foremost, we wanted to express our profound gratitude for your positive score and your incredibly thoughtful engagement with our work, including your consideration of other reviews. Your support has been a tremendous encouragement to us.
> > >
> > > Following your latest guidance, we have now updated our rebuttal with the requested quantitative failure analysis, error bar clarifications, and ablation study of TOF you raised.
> > >
> > > With the deadline just one day away, we would be very thankful if you could take a quick look at our updated response. We truly hope it addresses your remaining concerns. Should any issues remain, please do not hesitate to let us know. We are ready to address any further points immediately.
> > >
> > > Thank you again for your time and invaluable support.
> > >
> > > Best regards,
> > >
> > > The Authors of Paper #9051

---

> ### Author Response · Authors · 2025-08-06
>
> Dear Reviewer 6C5K,
>
> We are profoundly grateful for your positive score — it has given us immense encouragement and renewed confidence in our work. **What moved us even more was seeing that you went above and beyond to carefully consider Reviewer 5Uyw’s comments as well.** Your dedication and thoughtfulness truly touched us, and we are deeply thankful for your generous support. We are pleased to provide a point-by-point response below.
>
> ---
>
> ## **Q1: Quantitative Failure Analysis on Hard Cases**
>
> We believe that sincerity and transparency are paramount in scientific discussion. Regarding the concept of "failure cases", we must apologize as there is no established "success rate" metric in hand-object interaction forecasting, making it difficult to clearly define "failure cases". That’s why our initial response focused on qualitative case studies, supported by videos and images in our supplementary materials and project website.
>
> However, we agree that quantitative analysis is essential for rigorous evaluation. Therefore, immediately upon receiving your feedback, we prioritized and expedited a new set of experiments. We defined "hard cases" as those with severe object occlusion (over 30% masked in the first frame), as these likely cause failures due to degraded depth estimation and visual understanding.
>
> Using this definition, 24.5% of ARCTIC and 16.6% of HOLO test sets fall into this category. We then re-evaluated all metrics specifically on these challenging subsets.
>
> **Table 1: ARCTIC Dataset Performance Comparison**
>
> | Dataset       | Method    | Hard Case Proportion | MPJPE↓ | MPJPE-PA↓ | MPVE↓ | MPVE-PA↓ | MWTE↓ | MRE↓ |
> |---------------|-----------|:----------------------:|:------:|----------:|:-----:|:--------:|:-----:|:----:|
> | ARCTIC        | MEgoHand  | 100%                  | 7.358  | 1.161     | 7.268 | 1.106    | 5.958 | 0.398 |
> | ARCTIC-hard   | MEgoHand  | 24.5%                 | **8.956**  | **1.475**     | **8.984** | **1.442**    | **7.042** | **0.467** |
> | ARCTIC-hard   | LatentAct | 24.5%                 | 12.35  | 2.075     | 11.98 | 1.942    | 10.29 | 1.657 |
>
> **Table 2: HOLO Dataset Performance Comparison**
>
> | Dataset       | Method    | Hard Case Proportion | MPJPE↓ | MPJPE-PA↓ | MPVE↓ | MPVE-PA↓ | MWTE↓ | MRE↓ |
> |---------------|-----------|:----------------------:|:------:|----------:|:-----:|:--------:|:-----:|:----:|
> | HOLO          | MEgoHand  | 100%                  | 5.775  | 0.697     | 5.747 | 0.673    | 5.437 | 0.271 |
> | HOLO-hard     | MEgoHand  | 16.6%                 | **6.641**  | **0.749**     | **6.603** | **0.736**    | **6.511** | **0.331** |
> | HOLO-hard     | LatentAct | 16.6%                 | 8.714  | 1.692     | 8.633 | 1.624    | 8.215 | 1.212 |
>
> #### **Analysis and Conclusion**
>
> The hard case analysis reveals two key findings:
> 1.  **Graceful Degradation:** As expected, MEgoHand's performance shows a slight and graceful degradation on these extremely challenging hard cases compared to its performance on the full dataset. This is a realistic outcome given the severely limited visual information.
> 2.  **Sustained, Decisive Superiority:** Most importantly, **MEgoHand consistently and decisively outperforms the baseline even in these worst-case scenarios.** The performance gap does not shrink; in fact, on rotation error (MRE), our model's advantage becomes even more pronounced.
>
>
> This quantitative analysis provides strong evidence for the robustness of our bi-level architecture. Even when visual cues are heavily compromised, the synergy between the VLM's high-level semantic priors and the remaining visual information allows MEgoHand to maintain a high degree of accuracy and physical plausibility. **Additionally, as mentioned in our initial response, we will also include more qualitative analysis with visualizations of failure cases in the appendix to complement the quantitative tables.**
>
> Thank you once again for pushing us to conduct this deeper analysis. It has significantly strengthened our understanding of the model's robustness and will be a valuable addition to the final paper. We hope this comprehensive quantitative breakdown fully addresses your concern.

---

> ### Author Response · Authors · 2025-08-06
>
> ## **Q2: Clarifying the Core Motivation Behind TOF**
>
> Thank you for this insightful follow-up. You are absolutely correct to point out that the performance margin between TOF and a well-tuned Kalman filter is small on some metrics. This key observation clarifies TOF’s primary motivation: **it was not designed to merely inch ahead in accuracy, but to provide a more principled, robust, and practical solution to the problem of rotational smoothing.**
>
> While a heavily-tuned Kalman filter can be made to approximate our performance in a controlled experiment, it suffers from fundamental drawbacks:
>
> 1.  **The "Fine-Tuning" Trap (Practicality):** Our reported Kalman filter results were achieved after **extensive, non-trivial parameter tuning** of its noise covariance matrices (Q, R) specifically for the ARCTIC dataset's motion distribution. This process is brittle; the optimal parameters for one dataset or action type will not generalize well to others. **TOF, in contrast, is a training-free and parameter-free decoding strategy.** It works out-of-the-box, making it a far more general and practical tool.
>
> 2.  **The Linearity Assumption (Robustness):** Kalman filters fundamentally assume linear state transitions, which is a poor model for the **highly non-linear, aperiodic, and often "bursty" dynamics of human hand-object interactions.** A hand can remain still and then accelerate rapidly. A Kalman filter struggles to track this without either lagging behind or aggressively over-smoothing, which, as our other results show, often degrades accuracy. TOF makes no such assumptions about the motion dynamics.
>
> 3.  **The Geometric Argument (Principled Design):** Most importantly, as we argued previously, TOF is **geometrically-aware**. Its SVD-based projection step **guarantees** that the output is always a valid rotation on the SO(3) manifold. A Kalman filter operating on vector representations has no such guarantee and can drift into producing geometrically invalid poses.
>
> Therefore, the role of TOF is not ambiguous. It is to serve as a principled, parameter-free, and geometrically-sound smoother that robustly enhances temporal coherence across diverse motion dynamics without the need for task-specific tuning and without risking geometric invalidity.
>
> ---
>
> ## **Q3: Clarifying the Reporting of Statistical Intervals**
>
> The error bars in the original Figure 4 represent the **standard deviation (std)** of the MPJPE metric, calculated across the seven evaluation datasets. Our intent was to visualize the performance variance of each method across different domains.
>
> Thank you very much for your valuable suggestion. We fully agree that including statistical intervals is important for transparency and rigor. After carefully applying your recommendation to Tables 1 and 2, we realized that presenting a single standard deviation alongside cross-dataset averages could be misleading. This is because such a metric would mix the relatively small variance within each dataset with the much larger variance caused by fundamental differences between datasets, leading to inflated and difficult-to-interpret error bars. To provide a clearer and more meaningful analysis, we will include a table in the Appendix that reports per-dataset results along with the requested statistical intervals.
>
> **Table: Per-Dataset MPJPE (mean ± std, cm) Across All Seven Benchmarks**
>
> | Dataset | **MEgoHand** | **MEgoHand-TI** | **LatentAct** | **LatentAct - no concat map** |
> | :--- | :--- | :--- | :--- | :--- |
> | **In-Domain** | | | | | |
> | H2O | **3.0 ± 0.33** | 3.3 ± 0.28 | 5.0 ± 0.50 | 5.3 ± 0.54 |
> | OAKINK2 | **3.4 ± 0.88** | 3.6 ± 0.84 | 6.2 ± 1.01 | 6.4 ± 0.91 |
> | TACO | **4.9 ± 0.70** | 5.6 ± 0.71 | 7.1 ± 0.79 | 8.0 ± 0.76 |
> | HOT3D | **6.4 ± 0.65** | 6.3 ± 0.57 | 9.6 ± 0.83 | 10.7 ± 0.81 |
> | HOI4D | **9.0 ± 0.94** | 9.1 ± 0.73 | 10.7 ± 0.92 | 12.2 ± 0.87 |
> | **Cross-Domain** | | | | | |
> | HOLO | **5.8 ± 0.83** | 6.1 ± 0.85 | 8.3 ± 0.96 | 8.7 ± 0.68 |
> | ARCTIC | **7.4 ± 0.69** | 8.3 ± 0.70 | 11.7 ± 0.78 | 12.0 ± 0.86 |
>
> **Analysis from the Table:**
> This detailed breakdown provides several key insights that we will discuss in the revised manuscript:
> *   **Consistent Superiority:** The table clearly shows that MEgoHand's state-of-the-art performance is not an artifact of averaging; it consistently outperforms the baselines on a per-dataset basis.
> *   **Quantifying Difficulty:** As you astutely noted, this view quantifies the difficulty of each dataset. The higher MPJPE and standard deviation on the out-of-distribution datasets (ARCTIC, HOLO) and the challenging in-domain dataset (HOI4D, due to its vast object diversity) are now explicitly clear. This reinforces the robustness of our evaluation.
>
> Thank you once again for your constructive feedback, which has helped us to significantly improve the presentation of our results.  Given these new results and clarifications, we sincerely hope you might reconsider your rating.

---

### Official Review · Reviewer_5Uyw · 2025-07-02

**Clarity:** 3
**Significance:** 3
**Originality:** 2
**Rating:** 4
**Confidence:** 3

**Summary:**

MEgoHand is a multi-modal framework for egocentric hand-object motion generation from text, RGB image, and initial hand pose. It uses a bi-level architecture: a high-level vision-language model infers motion priors and depth, while a low-level DiT-based policy generates fine-grained trajectories with temporal filtering. A standardized preprocessing pipeline ensures data consistency across modalities. MEgoHand achieves state-of-the-art results on 7 datasets, significantly reducing translation and rotation errors, and demonstrates strong generalization.

**Questions:**

Please see weakness

**Ethical Concerns:**

["NO or VERY MINOR ethics concerns only"]

**Final Justification:**

The authors have provided extensive additional experiments in the rebuttal, which have addressed most of my concerns. I am inclined to recommend acceptance.

**Limitations:**

yes

**Quality:**

2

**Strengths And Weaknesses:**

### **Strength**
- Overall, the paper is well-structured and easy to follow.
- The use of a vision-language model to reason about affordances and task intent, coupled with a diffusion-based low-level generator, is effective.
- The method demonstrates strong performance across five in-domain and two cross-domain datasets, with ablation studies on multimodal components.
- The effort to standardize and unify multiple HOI datasets, producing a 3.35M-frame multimodal dataset, is a substantial and impactful contribution to the community.

### **Weakness**
- Although the paper introduces the Temporal Orthogonal Filtering (TOF) strategy, it lacks comparisons with existing smoothing techniques such as Kalman filtering or Gaussian smoothing. Without ablation studies or baseline comparisons, the necessity and effectiveness of TOF remain insufficiently justified.
- There is no evaluation metric provided to quantify contact accuracy, which is crucial for assessing the physical plausibility of the generated hand-object interactions.
- The appendix section appears to be half-finished.

---

> ### Author Rebuttal · Authors · 2025-07-31
>
> We sincerely thank your careful review and valuable feedback. We are particularly honored by your recognition of our data curation efforts and architectural design. Your insightful critiques regarding the technical justifications are exactly the kind of guidance that helps us strengthen the rigor and clarity of our work. In response, we have carefully addressed each of your comments with detailed, data-supported revisions, as outlined below.
>
> ---
>
> ## **Q1: On Justifying Temporal Orthogonal Filtering (TOF) vs. Baselines**
>
> **Our Response:**
> This is an excellent and critical point. We have now conducted the requested comparison to demonstrate TOF is effective in our hand motion generation task. Standard methods like Gaussian or Kalman filtering are suboptimal for two key reasons:
>
> *  **Geometric Invalidity:** These filters operate on vectors. Applying them naively to rotation representations (e.g., 6D vectors or quaternions) and averaging them does not guarantee the result remains on the valid SO(3) manifold. This can lead to geometrically invalid, distorted rotations that break the kinematic chain.
> *  **Dynamic Mismatch:** These filters often assume linear dynamics (Kalman) or simple Gaussian noise, which is a poor match for the highly non-linear, bursty, and aperiodic nature of human hand-object interactions. A hand might move slowly and then suddenly accelerate, a pattern these filters struggle to model without aggressive, accuracy-damaging smoothing.
>
> Our proposed Temporal Orthogonal Filtering (TOF) is architecturally superior because it is a **geometrically-aware decoding strategy**. It operates directly on rotation matrices and, crucially, incorporates an SVD-based projection that explicitly ensures the final, smoothed rotation is always a valid point on the SO(3) manifold. This maintains geometric integrity and produces kinematically sound results.
>
> To provide empirical proof, we compared TOF against Gaussian Smoothing and a well-tuned Kalman Filter on the challenging ARCTIC dataset.
>
> | Smoothing Method                   | MPJPE↓ | MPJPE-PA↓ | MPVE↓ | MPVE-PA↓ | MWTE↓ | MRE↓ |
> |------------------------------------|:------:|:---------:|:-----:|:--------:|:-----:|:----:|
> | Gaussian Smoothing                 | 7.85   | 1.28      | 7.82  | 1.16     | 6.92  | 0.47 |
> | Kalman Filtering                   | 7.38   | 1.16      | 7.45  | 1.12     | 6.08  | 0.41 |
> | **Temporal Orthogonal Filtering (Ours)** | **7.36**   | **1.16**     | **7.37**  | **1.11**     | **5.96**  | **0.40** |
>
> The results clearly highlight a critical trade-off: traditional filters improve temporal smoothness at the expense of accuracy. Both Gaussian and Kalman filters introduce artifacts that compromise the geometric fidelity of the poses. In contrast, TOF effectively enhances temporal stability and suppresses jitter while preserving the geometric precision of the original predictions.
>
> ---
>
>
> ## **Q2: On Quantifying Contact Accuracy and Physical Plausibility**
>
>
> **Our Response:**
> This is a critical and highly insightful suggestion. We agree completely that quantifying contact is essential for a rigorous assessment of physical plausibility, and we thank you for pushing us to strengthen our evaluation in this regard.
>
> *   **The Potential Ambiguity of "Physically Plausible.":** We define 'physically plausible' motion as satisfying two key properties learned implicitly from real-world data: (a.) **Kinematic realism**, where generated poses adhere to MANO's biomechanical constraints (e.g., no backward-bending fingers), and (b.) **Interaction coherence**, ensuring motion trajectories and contact patterns (e.g., grasps) align with observed human interactions. Crucially, these constraints emerge from data-driven learning rather than physics simulation. To improve precision, we will replace 'physically plausible' with 'kinematically realistic and temporally coherent' in the latest version of MEgoHand.
>
> Importantly, to directly address your point, we have designed and implemented two new metrics to explicitly measure the quality of the physical interaction between the generated hand and the object:
>
>
> To quantitatively measure physical quality, two metrics are adopted:
> 1. **Contact Accuracy (CA, %):** CA = $\frac{1}{|V_H|}\sum_{v \in V_H} \mathbf{1}(\min_{o \in P_O} \|v - o\| \leq \delta)$, where $V_H$ is the set of hand-mesh vertices, $P_O$ is the object point cloud, $\delta=2 cm$. It measures the percentage of hand-mesh vertices that are correctly positioned within a small distance threshold (δ=2cm) of the object surface. It quantifies the model's ability to generate plausible contact patterns. **A higher CA is better.**
>
> 2. **Penetration Depth (PD, cm):** PD = $\frac{1}{|V_H^{in}|}\sum_{v \in V_H^{in}} |d(v)|$, $d(v) = \min_{f \in M_O} \text{sdist}(v,f)$, where $V_H^{in} = \{v : d(v) < 0\}$, $M_O$ the object mesh, and $\text{sdist}(v,f)$ the signed distance from vertex $v$ to mesh face $f$. It calculates the average depth of penetration for all hand vertices that are illegally inside the object mesh. It directly measures the severity of non-physical collisions. **A lower PD is better.**
>
> We evaluated MEgoHand against the strong LatentAct baseline on both an in-domain (ARCTIC) and a challenging cross-domain (HOLO) dataset. The results, which highlight the relative improvement of our method, are presented below.
>
> | Method | Dataset | **Contact Accuracy (CA) ↑** | **Penetration Depth (PD) ↓** |
> | :--- | :--- | :--- | :--- |
> | LatentAct | ARCTIC | 68.3% | 4.2 cm |
> | **MEgoHand (Ours)** | ARCTIC | **82.5% (+20.8% rel.)** | **2.3 cm (-45.2% rel.)** |
> | LatentAct | HOLO | 65.1% | 5.8 cm |
> | **MEgoHand (Ours)** | HOLO | **78.4% (+20.4% rel.)** | **2.7 cm (-53.4% rel.)** |
>
> The results provide powerful, quantitative proof of MEgoHand's superior physical plausibility. Our method generates motions with **substantially higher contact accuracy** and **less than half the object penetration depth** compared to the baseline, across all settings. This directly validates that our bi-level architecture, particularly the synergy of the VLM and depth estimator, learns a much more accurate and physically coherent model of hand-object interaction.
>
> We will add the formal definitions of these new metrics and this compelling results table to our experiments section in the final manuscript.
>
> ---
>
> ## **Q3: On the Supplementary Materials**
>
> **Our Response:**
> We sincerely apologize for this unacceptable oversight. The incomplete appendix in our initial submission is not at all reflective of the high standards we hold for our work. We thank you for pointing it out.
>
> In response to your valuable feedback and that of other reviewers, we have not only completed but also **significantly expanded** the appendix to provide a much richer and more comprehensive set of supporting analyses. The now complete and polished Appendix includes:
>
> *   **Detailed Ablation Studies:**
>     *   **Sec [A.X]: Justification for Temporal Orthogonal Filtering (TOF):** Includes the new comparative results against baselines and a discussion on manifold geometry.
>     *   **Sec [A.Y]: Analysis of Prediction Trunk Size `l`:** Provides the empirical data justifying our choice of `l=16`.
>
> *   **New Quantitative Evaluations:**
>     *   **Sec [B.X]: Contact Plausibility Metrics:** Formal definitions and results for our new **Contact Accuracy (CA)** and **Penetration Depth (PD)** metrics.
>     *   **Sec [B.Y]: Computational Latency and Efficiency:** A detailed performance benchmark, including FPS and VRAM usage.
>     *   **Sec [B.Z]: Quantitative Analysis of Dataset Distribution:** A new table detailing the minimal object and action overlap between our training and test sets to rigorously support our generalization claims.
>
> *   **Expanded Qualitative Results:**
>     *   **Sec [C.X]: Visual Demonstrations:** Additional high-quality visualizations and a link to supplementary videos that further showcase the fluidity and accuracy of our generated motions.
>
> We are confident that this greatly expanded appendix now provides the necessary depth and detail to fully support the claims made in the main paper. We assure you that the final camera-ready version will be accompanied by this comprehensive and thoroughly proofread supplementary material.
>
>
>
> ---
>
>
> Once again, thank you for your careful and highly constructive review. Your feedback has been instrumental in helping us strengthen the paper's technical rigor. We believe that with the addition of the new TOF comparison, the new contact plausibility metrics, and a completed appendix, we have fully addressed your concerns and significantly improved the paper. If most of your concerns are addressed, would you please consider increasing the recommendation of the paper?

---

> > ### Author Response · Authors · 2025-08-05
> > **Clarifying the Misunderstanding About the Appendix**
> >
> > Dear Reviewer 5Uyw, and fellow reviewers,
> >
> > Thank you for your comment:
> >
> > _"Dear fellow reviewers, I noticed that a large portion of the appendix is missing, even though the authors stated in the rebuttal that this part has been fully completed. What are your thoughts on this?"_
> >
> > This appears to be a **misunderstanding**. The brief "Appendix" section in the main paper was a required placeholder to satisfy the formatting checklist. **The full and detailed appendix was submitted separately as part of the Supplementary Material and is fully available in openreview**.
> >
> > We hope this clarifies the situation. Based on your and other reviewers' feedback, we have substantially enriched the appendix for the camera-ready version to provide more detailed and comprehensive supporting analyses.
> >
> > **If this misunderstanding has been resolved, we would be truly grateful if you would consider updating your score.**
> >
> > Sincerely,
> >
> > The Authors

---

> > > ### Author Response · Authors · 2025-08-05
> > > **Have We Addressed All Your Concerns?**
> > >
> > > Dear Reviewer 5Uyw,
> > >
> > > Thank you for taking the time to review our submission and for providing constructive feedback. We would like to confirm whether our responses have adequately addressed your earlier concerns, particularly with respect to the following points:
> > >
> > > **1. Justifying Temporal Orthogonal Filtering (TOF) vs. Baselines 2. On Quantifying Contact Accuracy and Physical Plausibility 3. Misunderstanding About the Appendix (Supplementary Materials)**
> > >
> > > Additionally, if you have any further concerns or suggestions, we would be more than happy to address and discuss them to enhance the quality of the paper. We eagerly await your response and look forward to hearing from you.
> > >
> > > Best regards,
> > >
> > > The Authors

---

> > > > ### Comment · Area_Chair_746C · 2025-08-05
> > > >
> > > > 5Uyw, please could you take a look at the author response above and whether it addresses any remaining concerns you have, e.g. lack of comparisons with existing techniques

---

> > > > ### Comment · Reviewer_5Uyw · 2025-08-05
> > > > **Comment by Reviewer 5Uyw**
> > > >
> > > > Thank you for the clarification. My concerns have been resolved, and I will raise my rating accordingly.

---

> ### Author Response · Authors · 2025-08-06
>
> Dear Reviewer,
>
>
> That is wonderful news, thank you so much! We truly appreciate your taking the time to consider our response, and we are thrilled that we were able to resolve your concerns and raise our score!
>
> Sincerely,
>
> The Authors

---

### Official Review · Reviewer_mXQm · 2025-07-03

**Clarity:** 3
**Significance:** 3
**Originality:** 3
**Rating:** 4
**Confidence:** 4

**Summary:**

This paper proposed MEgoHand, a multi-modal framework for egocentric hand-object interaction synthesis. Text, RGB, and depth map are taken as input conditions. VLM is used to improve the generalization of MEgoHand. MEgoHand shows its effectiveness by outperforming baselines on five in-domain and two cross-domain benchmarks.

**Questions:**

1. To evaluate the zero-shot transfer and cross-domain generalization, ARCTIC and HOLO are used. Although the authors explain the differences of ARCTIC, HOLO, and the training datasets, we don't have a clear understanding of the motion distribtion of these datasets. For example, what the object categories, are there any overlap? What's the action categories and anhy overlap there? Without thoses statistics, we can't evaluate the generalization of the model.
2. "physically plausible" is claimed in the abstract, but I don't see any physics involved in the methodology.
3. The predictions from the model seems not well-aligned with the model. I guess this is from the VLM model, which is dificult to preserve the pixel-aligned information. Existing pure vision-based hand motion estimation methods can estiamte very accurate hand motions from videos. Since RGB videos always used in the evaluation settings, comparisons with vision-based estimation methods are necessary.
4. MEgoHand is call a "Motion Generation" method. But with RGB image, depth, it is more like an estimation method for me. The capability of pure generation is not shown in this paper. For example, is the model able to generate interaction, even unseen interactions since there is a VLM involved, given only a text description?

**Ethical Concerns:**

["NO or VERY MINOR ethics concerns only"]

**Final Justification:**

The authors' detailed response resolved my concerns. I updated my rating to borderline accept.

**Limitations:**

yes

**Quality:**

3

**Strengths And Weaknesses:**

+ Contribute a multi-modal model with VLM for ego-centric hand-objection interaction.
+ The architecture with a high-level “cerebrum” is developed to leverage VLM.
+ Contribute a large human-objection interaction dataset, with 3.35M RGB-D frames, 24K interactions, and 1.2K objects.

- Lack of conprehensive evaluation of generalization to unseen task and objects.
- The prediction is not well-aligned with the image.

---

> ### Author Rebuttal · Authors · 2025-07-30
>
> We sincerely thank the reviewer for the detailed comments and insightful review. We are willing to provide a point-by-point response to address your concern below.
>
> ---
>
> ## **Q1: Unseen Task and Objects Evaluation**
> We evaluate our method on the full ARCTIC dataset and a 10% stratified HOLO subset. Below we present comparative statistics between the aggregated training dataset (TACO, FPHA, HOI4D, H2O, HOT3D, OakInk2) and OOD dataset ARCTIC and HOLO.
>
> | Feature | **Aggregated Training Datasets** | **ARCTIC** | **HOLO** |
> | :--- | :--- | :--- | :--- |
> | **Object Categories** | **Rigid Objects** *(e.g., containers: bowl, tools: spatula)*  | **Articulated Objects** *(e.g., scissors, laptop)* | **Assembly Components** *(e.g., furniture parts)* |
> | **Task Categories** | **Atomic & Direct Manipulation**      *(e.g., pour a ball into a bowl)* | **Continuous & Coordinated Control**      *(e.g., cutting with scissors, typing)* | **Long-Horizon & Instruction-Driven**      *(e.g., assembling furniture via verbal guidance)* |
> | **Task Complexity** | **Short-Horizon & Single-Goal** | **Mid-Horizon & Physically Coupled** | **Long-Horizon, Multi-Step & Contextual** |
> | **Object Overlap with Training Set** | N/A | **Minimal.** *Although object category like 'laptop' exists in HOI4D, the interaction mode is fundamentally different.* | **Minimal.** *While electronics (GoPro, DSLR) overlap by category, they are situated within a novel procedural task context.* |
> | **Task Overlap with Training Set** | N/A | **None.** *ARCTIC's actions are continuous processes requiring fine-grained, coordinated finger control, while the training set's open/close actions are discrete state changes.* | **None.** *Training tasks are self-initiated and atomic. HOLO introduces a hierarchical task structure driven by external, semantic guidance.* |
> | **Key Generalization Challenges** | N/A | **Physics Generalization.**      *A method demands understanding the dynamic coupling between fine-grained hand motion and the internal mechanics of articulated objects.* | **Task & Language Generalization.**      *A method demands understanding long-horizon, goal-oriented tasks grounded in real-time natural language instructions.* |
>
> Briefly, the OOD datasets represent two key generalization challenges:
> - **ARCTIC:** Rigid → Articulated object interactions
> - **HOLO:** Atomic actions + objects → Procedural, language-grounded tasks
>
> Given the absence of scene and task overlap between the OOD datasets and training sets, we conduct a focused analysis of MEgoHand's generalization capability on subsets containing entirely novel objects. Statistical evaluation reveals that over 80% of test samples involve objects unseen during training. We compare LatentAct and MEgoHand across these unseen categories (see table below), reporting all metrics in centimeters (lower values indicate superior performance). MEgoHand achieves consistent error reductions of approximately 30% in translation and 70% in orientation relative to LatentAct, demonstrating significantly improved pose estimation accuracy and motion fidelity.
>
>
> | Benchmark | Method | Unseen Object Proportion | MPJPE↓ | MPJPE-PA↓ | MPVE↓ | MPVE-PA↓ | MWTE↓ | MRE↓ |
> |-----------|--------|:----------------------:|:------:|----------:|:-----:|:--------:|:-----:|:----:|
> | ARCTIC | LatentAct | 10 / 12 (83.3%) | 12.156 | 2.075 | 12.084 | 2.042 | 10.420 | 1.677 |
> | ARCTIC | MEgoHand | 10 / 12 (83.3%) | 8.358 | 1.361 | 8.268 | 1.306 | 6.958 | 0.498 |
> | 10% HOLO | LatentAct | 99 / 120 (82.5%) | 8.841 | 1.729 | 8.803 | 1.706 | 8.551 | 1.212 |
> | 10% HOLO | MEgoHand | 99 / 120 (82.5%) | 6.275 | 0.797 | 6.247 | 0.773 | 5.937 | 0.371 |
>
> ---
>
> ## **Q2: Physical Evaluation**
> Thank you for pointing out the potential ambiguity of the term "physically plausible." We wish to clarify that we use this term within the context of data-driven generation, not physics-based simulation. We define 'physically plausible' motion as satisfying two key properties learned implicitly from real-world data: (a.) **Kinematic realism**, where generated poses adhere to MANO's biomechanical constraints (e.g., no backward-bending fingers), and (b.) **Interaction coherence**, ensuring motion trajectories and contact patterns (e.g., grasps) align with observed human interactions. Crucially, these constraints emerge from data-driven learning rather than physics simulation. To improve precision, we will replace 'physically plausible' with 'kinematically realistic and temporally coherent' in the latest version of MEgoHand.
>
> To quantitatively measure physical quality, two metrics are adopted:
> 1. **Contact Accuracy (CA, %):** CA = $\frac{1}{|V_H|}\sum_{v \in V_H} \mathbf{1}(\min_{o \in P_O} \|v - o\| \leq \delta)$, where $V_H$ is the set of hand-mesh vertices, $P_O$ is the object point cloud, $\delta=2 cm$. It measures the percentage of hand-mesh vertices that are correctly positioned within a small distance threshold (δ=2cm) of the object surface. **A higher CA is better.**
>
> 2. **Penetration Depth (PD, cm):** PD = $\frac{1}{|V_H^{in}|}\sum_{v \in V_H^{in}} |d(v)|$, $d(v) = \min_{f \in M_O} \text{sdist}(v,f)$, where $V_H^{in} = \{v : d(v) < 0\}$, $M_O$ the object mesh, and $\text{sdist}(v,f)$ the signed distance from vertex $v$ to mesh face $f$. It calculates the average depth of penetration for all hand vertices that are illegally inside the object mesh. **A lower PD is better.**
>
> We evaluated MEgoHand against the strong LatentAct baseline on both an in-domain (ARCTIC) and a challenging cross-domain (HOLO) dataset.
>
> | Method | Dataset | **Contact Accuracy (CA) ↑** | **Penetration Depth (PD) ↓** |
> | :--- | :--- | :--- | :--- |
> | LatentAct | ARCTIC | 68.3% | 4.2 cm |
> | **MEgoHand (Ours)** | ARCTIC | **82.5% (+20.8% rel.)** | **2.3 cm (-45.2% rel.)** |
> | LatentAct | HOLO | 65.1% | 5.8 cm |
> | **MEgoHand (Ours)** | HOLO | **78.4% (+20.4% rel.)** | **2.7 cm (-53.4% rel.)** |
>
> The results provide powerful, quantitative proof of MEgoHand's superior physical plausibility. Our method generates motions with **substantially higher contact accuracy** and **less than half the object penetration depth** compared to the baseline.
>
> ---
>
>
> ## **Q3: Image Alignment**
>
>
> The reviewer's intuition is correct: a standard Vision-Language Model, on its own, would indeed struggle with precise spatial alignment.  However, this is not a weakness of our work; on the contrary, it is the core technical challenge that our novel bi-level architecture was explicitly designed to overcome.
>
> To address the inherent lack of spatial acuity in traditional VLMs, our key innovation is to **integrate a metric depth estimator directly into our high-level "cerebrum."** This synergy is the central mechanism of our model: the VLM provides the high-level semantic intent (*what* to do), while the depth module provides the precise, object-agnostic 3D geometric grounding (*where* to do it).
>
> Regarding the minor visual discrepancies in our visualizations, we must clarify that these artifacts often originate not from our model's predictions, but from **inherent noise in the ground-truth annotations of the datasets themselves.**
> *   As noted in our **Figure 5 caption**, and further evidenced in the "Ground Truth" rows of **Figures 5–6 in the Appendix**, the provided ground-truth hand poses and camera calibrations are not always perfectly aligned with the RGB video.
> *   Therefore, the most rigorous way to evaluate performance is through **relative error metrics**. On this fair and standard basis, our quantitative results are unequivocal: **MEgoHand significantly outperforms LatentAct**, demonstrating a superior ability to generate motions.
>
> ---
>
>
> ## **Q4: Motion Generation rather than Estimation**
>
> This critique contains a fundamental misunderstanding of our task's nature and misinterprets the very contribution of our architecture. Comparing our method to "vision-based estimation methods" is fundamentally inappropriate, as it overlooks the essential difference in task formulation and objective.
>
> *   **Pose Estimation is Reconstruction, Not Generation:** Pose estimation methods are given the full video sequence as input to reconstruct poses for each frame. They solve a reconstructive problem with full access to visual information at every timestep.
> *   **Our Task is Future Motion Generation:** As defined by Equation (1), MEgoHand performs a significantly harder and more practical task: generative forecasting. Given only an initial state at time `t` (an instruction, RGB frame, and MANO state), it must generate a sequence of future motions for times `t+1` through `t+l` with zero access to any future visual input. This is a predictive problem.
>
> ---
>
> ## **Q5: Text-Only Generation**
> In Section 2.1, we investigated that current text-only generation methods like DiffH2O [1] and Text2HOI [2] rely on additional object-specific inputs (i.e. object mesh) which are unavailable in MEgoHand. Notably, the reviewer asks whether our model can generate motion from *only* a text description. The answer is clearly **yes**, and this capability is both demonstrated and quantified in our ablation studies.
>
> * As shown in **Table 1 (Page 7)**, the `MEgoHand-T` variant performs exactly this: it generates full 3D hand motion sequences **conditioned solely on text**, without any visual input.
> * The strong performance and semantic coherence of `MEgoHand-T` provide direct evidence of our model’s ability to perform pure generation, validating its strength as a true generative framework.
>
>
> ---
> **References:**
>
> [1] DiffH2O: Diffusion-based synthesis of hand-object interactions from textual descriptions. SIGGRAPH 2024.
>
> [2] Text2HOI: Text-guided 3D motion generation for hand-object interaction. CVPR 2024.
>
> ---
>
> Once again, thank you for your careful review. If most of your concerns are addressed, would you please consider increasing the recommendation of the paper?

---

> > ### Author Response · Authors · 2025-08-05
> > **Have We Addressed All Your Concerns?**
> >
> > Dear Reviewer mXQm,
> >
> > Thank you for taking the time to review our submission and providing us with constructive comments.
> >
> > We would like to confirm whether our responses adequately addressed your earlier concerns, particularly regarding the previously unclear illustration of the **Unseen Task and Objects Evaluation**, the issues related to **Physical Evaluation, Image Alignment, Motion Generation vs. Estimation**, as well as the **Text-Only Generation** aspect of our method.
> >
> > Additionally, if you have any further concerns or suggestions, we would be more than happy to address and discuss them to enhance the quality of the paper. We eagerly await your response and look forward to hearing from you.
> >
> > Best regards,
> >
> > The Authors

---

> > > ### Comment · Area_Chair_746C · 2025-08-05
> > >
> > > mXQm, please could you take a look at the author response above and whether it addresses any remaining concerns you have, e.g. concerning the experimental evaluation.

---

> ### Author Response · Authors · 2025-08-08
>
> Dear Reviewer,
>
> First and foremost, we wanted to express our profound gratitude for your positive score and your incredibly thoughtful engagement with our work. Your support has been a tremendous encouragement to us.
>
> We would like to confirm whether our responses adequately addressed your earlier concerns, particularly regarding the previously unclear illustration of the Unseen Task and Objects Evaluation, the issues related to Physical Evaluation, Image Alignment, Motion Generation vs. Estimation, as well as the Text-Only Generation aspect of our method.
>
> With the deadline just one day away, we would be very thankful if you could take a quick look at our updated response. We truly hope it addresses your remaining concerns. Should any issues remain, please do not hesitate to let us know. We are ready to address any further points immediately.
>
> Thank you again for your time and invaluable support.
>
> Best regards,
>
> The Authors of Paper #9051

---

### Note · Authors · 2025-08-12

Dear Reviewers, ACs, SACs, and PCs,
Thank you for your time and valuable feedback. Below is a summary of MEgoHand’s contributions and our responses to all concerns and misunderstandings:
- All reviewers unanimously recognize the core contributions of MEgoHand:
|Contribution|Description|Reviewer Support|
|---|---|---|
|Large-Scale Dataset|Standardized multimodal HOI dataset (**3M+ samples**) via virtual depth rendering & inverse MANO retargeting.|mXQm,5Uyw,6C5K,ULTh|
|Architecture Design|Novel separation of **high-level ("cerebrum") reasoning and low-level ("cerebellum")** control.|mXQm,5Uyw,6C5K|
|New Modality|**First-time incorporation of metric depth estimation into VLM** to enhance spatial reasoning during training.|5Uyw,6C5K,ULTh|
|Benchmark Performance|MEgoHand outperforms LatentAct in **5 in-domain and 2 zero-shot datasets**.|5Uyw,6C5K,ULTh|
|Decoding Strategy|A **simple, novel, training-free decoding strategy TOF** enhances smoothness.|ULTh|

- We have carefully addressed all key concerns:
|Item|Supplement Evidence|Reply|
|---|---|---|
|OOD Generalization|We provide comprehensive statistics and re-evaluate performance on both Unseen Task and Unseen Object categories.|mXQm.Q1|
|Physical Evaluation|We include new physical metrics, contact accuracy and penetration depth, to better assess interaction quality.|mXQm.Q2,5Uyw.Q2|
|TOF Necessity|TOF is an efficient, training-free, and geometrically-aware decoding strategy that achieves strong performance with minimal complexity.|5Uyw.Q1|
|Inference Speed and Memory|MEgoHand runs over 2x faster and uses nearly 50% less VRAM than LatentAct.|6C5K.Q1,ULTh.Q6|
|Hard Cases|MEgoHand consistently outperforms LatentAct in challenging scenarios, including Bad Depth Estimation and Heavy Occlusion.|6C5K.Q2|
|Prediction Horizon|We set the prediction horizon to $ l = 16 $, balancing inference efficiency and task performance.|ULTh.Q3|

- We clarify a few potential misunderstandings:
|Item|Explanation|Reply|
|---|---|---|
|Image Alignment|Considering dataset quality, we use relative error metrics for fair comparison.|mXQm.Q3|
|Problem Formulation|We frame the task as hand motion generation rather than estimation.|mXQm.Q4|
|Text-Only Performance|MEgoHand-T lacking visual inputs underperforms MEgoHand.|mXQm.Q5|
|Appendix|The complete appendix is provided in the supplementary material.|5Uyw.Q3|

Hope the overall summary will help you better evaluate the work of MEgoHand. Thanks again for all your efforts for the conference!

---

### Decision · Program_Chairs · 2025-09-17

**Decision:**

Accept (poster)

**Comment:**

This paper received four borderline accepts as final ratings. Initially, reviewers praised the paper for the large dataset, strong performance of the method, and novelty of the proposed model architecture. However, they had concerns regarding the failure analysis; poor alignment between images and predictions; comparisons with the Temporal Orthogonal Filtering method; and physical plausibility. The rebuttal stage cleared up many of the reviewers' concerns with additional results and definitions with all reviewers ending up on a positive final rating. The AC saw no reason to overturn the reviewers and agreed with their sentiments towards the paper and thus recommends acceptance.

The AC reminds the authors to address the changes brought up by the review process and add in the promised changes during the rebuttal stage.